# SPARKLE: A ROBUST AND VERSATILE REPRESENTATION FOR POINT CLOUD BASED HUMAN MOTION CAPTURE

**Yiming Ren**[1,2]*, **Yujing Sun**[2]*, **Aoru Xue**[1], **Kwok-Yan Lam**[2]†, **Yuexin Ma**[1]†
[1]ShanghaiTech University, China
[2]Digital Trust Centre, Nanyang Technological University, Singapore
{renym2022,mayuexin}@shanghaitech.edu.cn

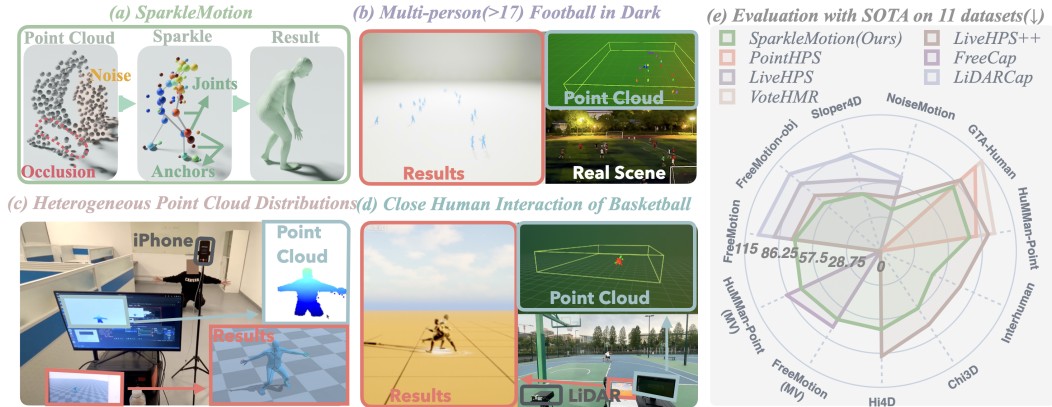

Figure 1. We introduce (a) *SparkleMotion*, a point-based real-time motion capture solution based on a novel human representation structure, *Sparkle*, benefiting both internal kinematics from skeletal joints and external geometry from surface anchors. It is capable of handling challenging scenarios in the real world, such as (b) multi-person football in dark, (c) heterogeneous point cloud distributions, and (d) close human interaction of basketball. The raw point cloud input and our real-time results are illustrated in the corresponding scenarios. **The real-time video is included in the supplementary materials**. *SparkleMotion*'s superiority to SOTA approaches and strong generalization across 11 datasets are showcased in (e), where methods don't work has area size 0, measured in Mesh Vertex Error($\downarrow$).

## ABSTRACT

Point cloud-based motion capture leverages rich spatial geometry and privacy-preserving sensing, but learning robust representations from noisy, unstructured point clouds remains challenging. Existing approaches face a struggle trade-off between point-based methods (geometrically detailed but noisy) and skeleton-based ones (robust but oversimplified). We address the fundamental challenge: how to construct an effective representation for human motion capture that can balance expressiveness and robustness. In this paper, we propose *Sparkle*, a structured representation unifying skeletal joints and surface anchors with explicit kinematic-geometric factorization. Our framework, *SparkleMotion*, learns this representation through hierarchical modules embedding geometric continuity and kinematic constraints. By explicitly disentangling internal kinematic structure from external surface geometry, *SparkleMotion* achieves state-of-the-art performance not only in accuracy but crucially in robustness and generalization under severe domain shifts, noise, and occlusion. Extensive experiments demonstrate our superiority across diverse sensor types and challenging real-world scenarios.

---

*Equal contribution.
†Correspondence authors.

## 1 INTRODUCTION

Human motion capture (MoCap) (Horberry et al., 2022; Ye et al., 2022; Xu et al., 2020; 2022; Jiang et al., 2023; Kaufmann et al., 2021; Yi et al., 2022; Charles et al., 2016; Belagiannis & Zisserman, 2017; Kim et al., 2019; Yi et al., 2021; Andriluka et al., 2018) has emerged as a fundamental technology for a wide range of human-centric applications, encompassing sports performance analysis, healthcare, virtual reality, human–robot interaction, etc. Although mainstream research on MoCap is based on a wide range of data modalities, point cloud based motion capture using LiDAR (Ren et al., 2024b;a; Dai et al., 2022) or depth cameras (Cai et al., 2023) stands out by offering distinct advantages, notably accurate depth perception and stronger guarantees of data privacy, outweighing wearable sensor-based (Yi et al., 2021; 2022) and camera-based (Shen et al., 2024; Shin et al., 2024) approaches. Furthermore, the rich geometric and spatial information inherent in point clouds enables accurate 3D localization and robust motion reconstruction, even under unconstrained conditions.

However, an everlasting challenge in point cloud-based MoCap is what intermediate representation should learn from unstructured, noisy, and often incomplete 3D point clouds. Existing approaches learn either (1) direct point-based representations (Qi et al., 2017; Vu et al., 2022), which retain geometric detail but lack structural priors, making them susceptible to noise and occlusion, or (2) skeleton-based abstractions (Ren et al., 2024b;a), which provide strong kinematic structural priors but discard the surface-level details essential for resolving rotational ambiguities and capturing nuanced shape, both categories struggling to achieve expressiveness and robustness simultaneously.

A powerful representation should tightly couple the robust structural priors of internal kinematics, which are resistant to noise and occlusion, with external geometry to preserve fine-grained details. However, a simple combination of existing skeletal and surface representations is insufficient, as each component itself suffers from inherent limitations: Skeleton-only methods lack surface constraints for recovering fine-grained details, while point-only methods are unstable under sparse or noisy observations. To this end, we propose a unified intermediate representation, *Sparkle*, that integrates internal kinematics with external surface geometry, thereby preserving skeletal coherence (robustness) while maintaining surface fidelity (expressiveness) across varying point cloud qualities and diverse scenarios. Crucially, the novelty of Sparkle lies not only in their union but also in the redesign of each constituent: (1) For the skeletal component, we introduce a *point-aligned* estimation process that explicitly models the spatial correspondence between the point cloud and the skeleton, moving beyond the common practice of direct regression, which often misaligns with the observed geometry. (2) For the surface component, we propose a set of semantically consistent *anchors* that are dynamically refined through a skeleton-guided, geometry-aware mechanism, offering a more stable and informative representation than unstructured points. This dual-novelty design ensures that *Sparkle* is a fundamentally more expressive and robust representation, rather than a trivial assemblage of existing ones.

To be more specific, building on *Sparkle*, we further propose **SparkleMotion**, a robust and effective point cloud-based human motion capture approach that advances both representation learning and practical MoCap performance. Firstly, we construct the **Sparkle** Representation, which is the unification of an internal kinematics representation and an external geometry representation. **Internal Kinematics Representation** consists of a set of accurately estimated skeletal joints. To obtain this, we design a *Point-aligned Skeleton Tracker* that learns to predict precise skeletal joints and global body translation by explicitly modeling spatial correspondences between point cloud surfaces and the target skeleton structure, while leveraging both local and global features through a segmentation-aware design. **Surface Geometric Representation**, in contrast, consists of a set of precise surface anchors predicted by a *Skeleton-guided Anchor Estimator*. This estimator learns to fuse the structural prior with the observed geometric evidence, predicting surface anchors based on predicted skeletal joints and learning a non-linear correction to a linear initial guess. To be clear, we first initialize the anchors by leveraging the structural relationships between internal joints and external surface points. Then, by integrating both skeletal and surface features, the estimator enhances the stability and robustness of anchor predictions. As shown in Figure 1, *Sparkle* unifies skeletal joints and surface anchors and injects structured priors to guide learning from point clouds. Then we design a *Sparkle-based SMPL Solver*. The Solver decomposes axis-angle poses into swing-twist components (Li et al., 2021) analytically by aligning each template bone to the observed joint vector and calculating the in-bone twist via anchor alignment. This initialization exploits Sparkle's fac-

torized kinematic–surface structure, after which human pose and shape can be conveniently refined with an optimization network.

Extensive experiments across a range of 11 challenging MoCap benchmarks demonstrate that *SparkleMotion* consistently outperform State-Of-The-Art point-based MoCap methods in terms of accuracy and robustness, even under multi-person large-scale scene, cross-sensor domain shifts, and occlusion and noise by close human interaction as shown in Figure 1. Furthermore, *SparkleMotion* generalizes effectively to data from various LiDARs and depth cameras, like iPhones, as well as multi-view capture setups, substantiating its utility as a scalable representation for real-world deployments. Leveraging our pipeline, we also develop a **real-time MoCap system** capable of stable and high-quality 3D human reconstruction in-the-wild, paving the way for broad-scale, privacy-aware, and interpretable human motion understanding. We summarize our contributions as follows:

- We propose *Sparkle*, a disentangled structured representation that unifies skeletal joints and surface anchors, explicitly designed to provide a stronger structural prior for learning from point clouds, achieving a promising balance between expressiveness and robustness.
- We develop *SparkleMotion*, a robust and generalizable MoCap framework that harnesses Sparkle representation for accurate, noise-resilient, and real-time motion capture.
- We establish new State-Of-The-Art performance on multiple public benchmarks, demonstrating *SparkleMotion*'s strengths in accuracy, robustness, cross-domain generalization, and scalability for real-world applications through comprehensive evaluations.

## 2  LITERATURE REVIEW

### 2.1  POINT CLOUD-BASED MOTION CAPTURE

Human motion capture has been widely explored using wearable sensors and vision-based methods. Wearable MoCap systems, such as inertial measurement unit-based (Ren et al., 2023; Yi et al., 2021; Huang et al., 2018a) and marker-based approaches (OptiTrack; Loper et al., 2014; Xu et al., 2019), offer high-quality motion reconstruction but require specialized hardware, limiting their flexibility and ease of use. To achieve a more non-intrusive solution, camera-based methods including monocular (Pons-Moll et al., 2014; Kanazawa et al., 2018; Alldieck et al., 2017; Shin et al., 2024; Wang et al., 2024; Sun et al., 2023) , multi-view (Dou et al., 2017; Huang et al., 2018b; Malleson et al., 2019), and RGB-D sensors (Zimmermann et al., 2018; Dang et al., 2023), estimate human motion from images or videos, leveraging deep learning to infer 3D human poses from visual cues. However, these methods are limited to light conditions, global localization, and privacy concerns.

Point cloud-based MoCap methods mainly leverage LiDAR and depth cameras to reconstruct human motion without relying on wearable markers or identity tracking. These approaches are particularly advantageous for privacy-sensitive applications and challenging environments. However, due to the distinct characteristics of LiDAR and depth camera point clouds, existing methods are often tailored to a specific sensor type, limiting their generalizability. LiDAR provides long-range depth perception but produces uneven and unstructured point distributions. LiDAR-based MoCap methods (Ren et al., 2024b; 2023; 2024a) are specifically designed for large-scale, sparse, and noisy LiDAR point clouds, where the data lacks rich geometric details. They extract skeletal structures from the limited point cloud input and focus primarily on optimizing the skeleton joints. Thus, when faced with the dense and detailed point clouds provided by depth cameras, they struggle to fully utilize the abundant geometric information, leading to suboptimal motion reconstruction. Conversely, depth camera-based MoCap (Cai et al., 2023; Liu et al., 2021) operates on dense, high-resolution point clouds, making it well-suited for indoor environments and detailed motion estimation. Depth sensors provide rich point clouds but suffer from limited range, occlusions, and noise. Methods designed for depth cameras focus on point cloud features to refine body part estimation but often lack robustness in outdoor settings or under sensor noise, making them ineffective when applied to sparse and irregular LiDAR data.

### 2.2  HUMAN MOTION REPRESENTATION

Traditional marker-based motion capture (Raskar et al., 2007; Loper et al., 2014) provides highly accurate, physically interpretable joint and marker trajectories, but requires elaborate hardware setup

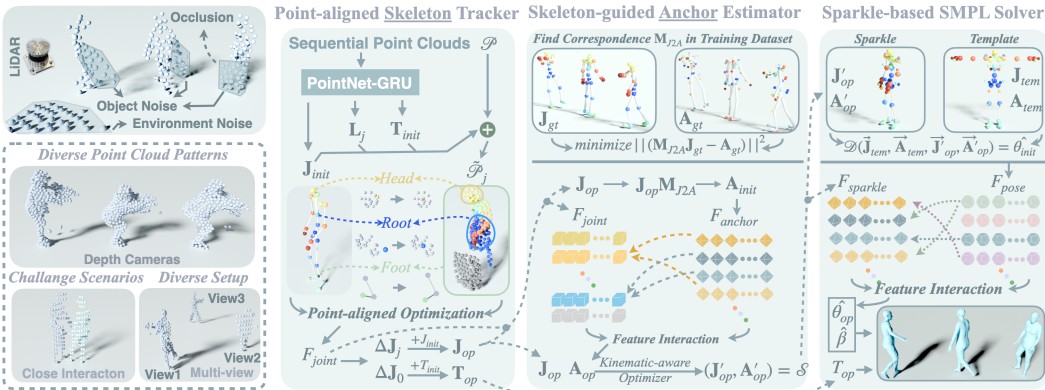

Figure 2: The pipeline of SparkleMotion. It can take point clouds of diverse patterns as input in different challenge scenarios, as shown on the left. SparkleMotion consists of three primary modules, the Point-aligned Skeleton Tracker, and Skeleton-guided Anchor Estimator construct the *Sparkle* Representation, and the Sparkle-based SMPL Solver for motion reconstruction.

and calibration, restricting its usability to controlled environments. IMU-based systems (Ren et al., 2023; Yi et al., 2021; Huang et al., 2018a) enable more flexible deployment by reconstructing internal joint positions from sparse sensor measurements. However, these approaches are sensitive to drift, misplacement, and rely on precise joint-to-sensor mappings. Vision-based algorithms have made significant strides by regressing 2D or 3D keypoints from single or multi-view images (Shin et al., 2024; Shen et al., 2024), sometimes utilizing virtual markers (Ma et al., 2023a) to encourage structural consistency. Despite impressive progress, monocular methods remain fundamentally limited by depth ambiguities and occlusion, and errors in 2D detection may cascade through the 3D reconstruction pipeline. Within the domain of point cloud-based MoCap, two primary representation strategies exist. Some methods operate directly on raw, unstructured point sets (Qi et al., 2017; Fan et al., 2023), which preserves geometric detail but often suffers from noise, occlusion, and lack of spatial regularity. Others extract skeleton-based abstractions (Ren et al., 2024b;a; Xue et al., 2025), focusing on stable joint estimation, but sacrificing detailed surface information crucial for full-body mesh recovery and subtle pose nuances. These observations highlight a key gap in current approaches: existing intermediate representations either prioritize structural consistency or surface fidelity, but rarely both. Addressing this, our work proposes a novel unified representation—*Sparkle*—that jointly encodes internal skeletal joints and external surface keypoints. This structured formulation bridges the gap between kinetic stability and geometric richness, enabling more robust, generalizable, and expressive motion capture from diverse point cloud data.

## 3 METHODOLOGY

The rich and unconstrained 3D spatial information offered by point clouds makes point cloud–based human motion capture highly attractive for real-world applications. Nonetheless, their unstructured, sparse, and often noisy nature poses significant challenges for learning consistent and robust human representations. To address this, we introduce a novel structured human representation, *Sparkle*, explicitly designed to bridge the gap between internal kinematic structure and external surface geometry, maintaining robustness and accuracy in diverse and challenging scenarios. As illustrated in Figure 2, *Sparkle* (Section 3.1) comprises an internal and an external representations: (1) Internal Kinematics representations, estimated by a Point-aligned Skeleton Tracker (PST) and (2) External Geometry Representation, optimized via a Skeleton-Guided Anchor Estimator (SAE). Finally, we present *Sparklemotion*, a Sparkle-based SMPL Solver (SSS) (Section 3.2) that predicts human shape and pose by regressing precise SMPL parameters from the unified Sparkle representation through an efficient geometric initialization and iterative refinement strategy. Please refer to Appendix A for details on model architecture.

**Problem Statement**  Let $\mathcal{P}_t \in \mathbb{R}^{256 \times 3}$ denote the input point cloud at frame $t$ after the farthest point sampling (Qi et al., 2017) and normalization. Notice that for the sake of clarity, we omit

the symbol for time $t$ below, but our input and output are all time series. Our goal is to estimate SMPL (Loper et al., 2015) parameters that define the predicted human mesh $\mathbf{V} \in \mathbb{R}^{6890 \times 3}$, including shape $\boldsymbol{\beta} \in \mathbb{R}^{10}$, pose $\boldsymbol{\theta} \in \mathbb{R}^{72}$ (in axis-angle form), and global translation $\mathbf{T} \in \mathbb{R}^3$. Notice that we extract 24 skeletal joints $\mathbf{J} \in \mathbb{R}^{24 \times 3}$ and 32 surface anchors $\mathbf{A} \in \mathbb{R}^{32 \times 3}$ from $\mathbf{V}$. And each joint rotation can be decomposed into swing $\mathbf{R}^{sw}$ and twist $\mathbf{R}^{tw}$ components as $\mathbf{R} = \mathbf{R}^{sw}\mathbf{R}^{tw}$.

## 3.1 SPARKLE REPRESENTATION

The design of the *Sparkle* representation is motivated by the observation that internal skeletal joints provide a robust structural prior but form an incomplete description of human pose, as they lack the surface details required to resolve rotational ambiguities. In contrast, external surface points preserve rich geometric information but are inherently vulnerable to noise and occlusion due to the absence of structural priors. To reconcile these complementary properties, we introduce *Sparkle*, a unified representation that combines an Internal Kinematics Representation 3.1.1 with a Surface Geometric Representation 3.1.2, thereby enhancing both expressiveness and robustness. In general, the *Sparkle* representation is defined as $\mathcal{S} = [\mathbf{J}'_{op}, \mathbf{A}'_{op}]$, which explicitly unifies 24 optimized skeletal joints $\mathbf{J}_{op}'$ as the Internal Kinematics Representation with 32 optimized anchors $\mathbf{A}_{op}'$ as the Surface Geometric Representation. Together, these components constitute a comprehensive encoding of human pose, as elaborated in the following.

### 3.1.1 INTERNAL KINEMATICS REPRESENTATION

The Internal Kinematics Representation is defined as a set of optimized skeletal joints. However, deriving a consistent skeletal structure from highly variable point clouds remains challenging. To address this, we propose the Point-aligned Skeleton Tracker (PST), which establishes explicit, learnable spatial correspondences between raw point clouds and skeletal joints through a segmentation-aware design, enabling robust joint estimation across diverse and noisy observations.

Given a point cloud $\mathcal{P}$, our PST module first employs a PointNet backbone with a bidirectional GRU to predict initial joint positions $\mathbf{J}_{\text{init}}$ and global translation $\mathbf{T}_{\text{init}}$. It then performs implicit semantic segmentation of $\mathcal{P}$ into body parts via a part-based decomposition, yielding point-wise labels $\mathbf{L}_j \in 0, 1, \dots, 24$ that associate each point with its corresponding joint. This segmentation establishes critical spatial correspondences that guide subsequent refinement. Subsequently, the skeleton joints are optimized based on a key insight: leveraging the predicted labels to establish correspondences between points and joints. To be clear, we first center the entire point cloud using the predicted global translation, $\mathcal{P}_{\text{centered}} = \mathcal{P} - \mathbf{T}_{\text{init}}$, Then, for each joint $j$, we extract the subset of points $\mathcal{P}_j = \{p \in \mathcal{P}_{\text{centered}} : \mathbf{L}_j(p) = j\}$ that share label $j$ from the centered point cloud. Finally, we normalize these points relative to their corresponding predicted joint, $\tilde{\mathcal{P}}_j = \{p - \mathbf{J}_{\text{init},j} : p \in \mathcal{P}_j\}$.

With above operations, the global regression problem is reformulated as a set of local refinement tasks. We develop the Point-aligned Optimization, enables the use of a lightweight, shared PointNet to process each local point set $\tilde{P}_j$ and get the local point feature $F_{\text{joint}}$, then use the MLP decoder to predict a residual offset $\Delta \mathbf{J}_j$. Sharing weights across joints compels the network to learn a generic refinement function that is invariant to specific body parts, thereby significantly enhancing generalization. To handle cases where a joint has too few associated points ($|\tilde{P}_j| < 3$), we enhance the feature by the local neighboring feature. This allows the model to propagate confident predictions to uncertain regions, a form of learned spatial reasoning.

After the refinement network predicts residual offsets $\Delta \mathbf{J}_j$ for each joint $j \in \{0, 1, \dots, 23\}$, the optimized joint positions and optimized translations (denoted as $\mathbf{J}_{\text{op}}$ and $\mathbf{T}_{\text{op}}$) are computed by adding these residuals to the initial estimates:

$$\mathbf{J}_{\text{op}} = \mathbf{J}_{\text{init}} + \Delta \mathbf{J}_j, \quad \mathbf{T}_{\text{op}} = \mathbf{T}_{\text{init}} + \Delta \mathbf{J}_0,$$

where $\Delta \mathbf{J}_j$ represents the predicted residual offset for joint $j$, enabling precise alignment with the underlying point cloud structure.

Finally, joint positions $\mathbf{J}_{\text{op}}$ are further optimized together with global translation, and point-wise classification via loss $\mathcal{L}_{\text{PST}}$

$$\mathcal{L}_{\text{PST}} = \lambda_1 \mathcal{L}_{\text{MSE}}(\mathbf{J}_{\text{op}}, \mathbf{J}_{\text{gt}}) + \lambda_2 \mathcal{L}_{\text{CE}}(\mathbf{L}_j, \mathbf{L}_{j_{\text{gt}}}) + \lambda_3 \mathcal{L}_{\text{MSE}}(\mathbf{T}_{\text{op}}, \mathbf{T}_{\text{gt}}), \tag{1}$$

where $\mathcal{L}_{\text{MSE}}$ is the Mean Squared Error loss, and $\mathcal{L}_{\text{CE}}$ is the cross-entropy loss for label classification, with $\lambda_1 = 1.0$, $\lambda_2 = 0.5$, and $\lambda_3 = 1.0$.

### 3.1.2 SURFACE GEOMETRIC REPRESENTATION

The surface geometry is represented by a set of surface anchors. Since directly predicting these anchors from raw point clouds is challenging due to their unstructured nature, we propose the Skeleton-guided Anchor Estimator (SAE). SAE adopts a prediction-with-initialization strategy, in which a strong initial estimate is first obtained and subsequently refined by incorporating geometric context.

**Structural Initialization.** We derive a set of 32 canonical anchors $\mathbf{A}_{\text{gt}} \in \mathbb{R}^{32 \times 3}$ from the SMPL mesh vertices via PCA (Ma et al., 2023b) to maximize their representational power. A linear mapping between internal joints and external anchors is precomputed from ground-truth data via Least Squares:

$$\mathbf{M}_{\text{J2A}} = (\mathbf{A}_{\text{gt}}^\top \mathbf{A}_{\text{gt}})^{-1} \mathbf{A}_{\text{gt}}^\top \mathbf{J}_{\text{gt}}, \tag{2}$$

which provides a fixed, geometry-agnostic prior $\mathbf{A}_{\text{init}} = \mathbf{J}_{\text{op}} \mathbf{M}_{\text{J2A}}$ that roughly places the anchors based on the predicted skeletal pose alone.

**Geometry-Aware Refinement.** The initial estimate $\mathbf{A}_{\text{init}}$ lacks detail and is vulnerable to errors in $\mathbf{J}_{\text{op}}$. We refine it by learning a non-linear correction that attends to the actual geometric feature $F_{\text{joint}}$. This is achieved through a cross-attention mechanism: joint features $F_{\text{joint}}$ from PST serve as queries ($Q$) that represent what we expect the local surface geometry to look like given the current pose. In parallel, a point cloud transformer processes the initial anchors $\mathbf{A}_{\text{init}}$ to generate the anchor feature $F_{\text{anchor}}$ as keys ($K$) and values ($V$), which represent what the actual local geometry is. The cross-attention operation allows each joint query to selectively attend to and aggregate information from the most relevant geometric features in the point cloud. This enables the model to learn to compensate for both errors in the initial joint estimates and the limitations of the linear model, effectively fusing the structural prior with observed geometric details.

Moreover, instead of predicting explicit confidence scores for each anchor, we leverage the learned anchor-based segmentation $\mathcal{L}_a$ of the point cloud to define anchor reliability. By analyzing the segmentation quality and point distribution around each anchor, we implicitly derive confidence measures that allow the downstream solver to weight the reliability of this learned representation. The entire estimator is trained with a loss that balances anchor accuracy:

$$\mathcal{L}_{\text{SAE}} = \lambda_4 \mathcal{L}_{\text{MSE}}(\mathbf{A}_{\text{op}}, \mathbf{A}_{\text{gt}}) + \lambda_5 \mathcal{L}_{\text{CE}}(\mathbf{L}_a, \mathbf{L}_{a_{\text{gt}}}) \tag{3}$$

where $\mathbf{L}_{a_{\text{gt}}}$ denotes the ground truth segmentation labels, $\lambda_4 = 1.0$ and $\lambda_5 = 0.5$.

### 3.1.3 JOINT REPRESENTATION

With the optimized skeleton joints $J_{op}$ (Section 3.1.1) and surface anchors $A_{op}$ (Section 3.1.2), we incorporate a final kinematic optimization step (Ren et al., 2024a) to enforce temporal consistency, yielding the final *Sparkle* representation $\mathcal{S} = [J'_{op}, A'_{op}]$ for the following SMPL solver.

## 3.2 SPARKLE-BASED SMPL SOLVER

At this point, we obtain the *Sparkle* representation $\mathcal{S}$, which explicitly factorizes the human motion into two complementary spaces: a kinematic space of joint configurations $\mathbf{J}'_{op}$ and a geometric space of surface deformations $\mathbf{A}'_{op}$. This design incorporates a powerful physical inductive bias, significantly reducing the learning complexity compared to unstructured representations. Each component specializes naturally: joints provide kinematic constraints for robust pose estimation, while anchors encode local surface details essential for resolving shape ambiguities.

We then propose Sparkle-based SMPL Solver(SSS) to leverage this factorization for efficient estimation of SMPL parameters of human shape and pose. Given the optimized representation $\mathcal{S}$, we recover body parameters $(\theta, \beta)$ through a two-stage process: a geometric initialization that exploits the separate constraints of each space, followed by a learned refinement that integrates both information streams for final precision.

### 3.2.1 GEOMETRY-DRIVEN INITIALIZATION

A key advantage of our structured representation is that it enables deterministic initialization through explicit geometric constraints, avoiding the biases inherent in learned initialization schemes.

This parameter-free procedure requires only: (1) bone vector composition $\vec{J} = J_2 - J_1$; (2) trigonometric operations on vectors (dot and cross products); and (3) Rodrigues' rotation formula.

For each bone $b$ defined in SMPL, we decompose axis-angle rotation into **swing-twist components** (Li et al., 2021) using associated joints and anchors ($\mathbf{J}_{\text{tem}}, \mathbf{J}'_{op}, \mathbf{A}_{\text{tem}}, \mathbf{A}'_{op}$), where $\mathbf{J}_{\text{tem}}$ and $\mathbf{A}_{\text{tem}}$ means the joint positions and anchor positions in the zero pose(T-Pose) of SMPL.

**Swing Rotation:** The swing component $\mathcal{D}(\alpha_{\text{sw}}[\vec{n}_{\text{sw}}]_\times)$ aligns the template bone direction $\vec{\mathbf{J}}_{\text{tem}}$ to the predicted direction $\vec{\mathbf{J}}'_{op}$ using only skeletal information:

$$\vec{n}_{\text{sw}} = \frac{\vec{\mathbf{J}}_{\text{tem}} \times \vec{\mathbf{J}}'_{op}}{\|\vec{\mathbf{J}}_{\text{tem}} \times \vec{\mathbf{J}}'_{op}\|}, \quad \alpha_{\text{sw}} = \arccos\left(\frac{\vec{\mathbf{J}}_{\text{tem}} \cdot \vec{\mathbf{J}}'_{op}}{\|\vec{\mathbf{J}}_{\text{tem}}\|\|\vec{\mathbf{J}}'_{op}\|}\right). \tag{4}$$

**Twist Rotation:** The twist component $\mathcal{D}(\alpha_{\text{tw}}[\vec{n}_{\text{tw}}]_\times)$ then rotates around the bone axis $\vec{n}_{\text{tw}} = \vec{\mathbf{J}}'_{op}/\|\vec{\mathbf{J}}'_{op}\|$ to align template anchors $\mathbf{A}_{\text{tem}}$ with predicted anchors $\mathbf{A}'_{op}$:

$$\alpha_{\text{tw}} = \arctan 2\left(\|\mathbf{A}_{\text{tem}} \times \mathbf{A}'_{op}\|, \mathbf{A}_{\text{tem}} \cdot \mathbf{A}'_{op}\right). \tag{5}$$

The complete rotation $R = R^{\text{sw}} R^{\text{tw}}$ is computed via Rodrigues' formula, providing a physically consistent initial pose estimate $\hat{\theta}_{\text{init}}$ without learned parameters. This demonstrates how *Sparkle*'s explicit structure facilitates geometrically meaningful computation.

### 3.2.2 SPARKLE-GUIDED PARAMETRIC REFINEMENT

The geometrically initialized pose $\hat{\theta}_{\text{init}}$ provides a strong, physically plausible starting point. However, this deterministic solution has inherent theoretical limitations that affect its stability and uniqueness. First, the swing-twist decomposition is susceptible to kinematic singularities. Specifically, the swing rotation becomes undefined when the template bone vector $\vec{\mathbf{J}}_{\text{tem}}$ and the observed bone vector $\vec{\mathbf{J}}_{op}$ are nearly aligned or anti-aligned, as their cross product approaches zero, making the rotation axis $\vec{n}_{\text{sw}}$ numerically unstable. Second, the solution for the twist component lacks uniqueness under ambiguous geometric evidence. The twist angle $\alpha_{\text{tw}}$ is estimated from the alignment of surface anchors, but this becomes ill-posed when anchors are occluded, noisy, or lie close to the bone axis (becoming co-linear), leading to multiple valid twist angles that satisfy similar geometric constraints. Consequently, the purely geometric formulation is sensitive to noise in the estimated *Sparkle* representation and may converge to sub-optimal local minima, particularly in the presence of occlusions or when the observed point cloud evidence is ambiguous. Furthermore, it cannot fully encapsulate the complex, non-linear correlations between body shape $\beta$, pose $\theta$, and the observed *Sparkle* features.

To bridge this gap between the analytical initialization and the final, refined mesh, we employ a lightweight cross-attention network to learn a non-linear correction. In this refinement stage, the initial pose parameters $\hat{\boldsymbol{\theta}}_{\text{init}}$ are encoded to get pose feature $F_{\text{pose}}$ as queries. The *Sparkle* features $F_{\text{sparkle}}$ serve as keys and values, representing the observed geometry feature. The cross-attention mechanism thus learns to adjust the initial pose by attending to the specific geometric constraints provided by the *Sparkle*, effectively performing a learned, iterative refinement on the plausible poses $\hat{\boldsymbol{\theta}}_{\text{op}}$ and shape $\hat{\boldsymbol{\beta}}$. The solver is trained with regression loss:

$$\mathcal{L}_{\text{SSS}} = \lambda_6 \mathcal{L}_{\text{MSE}}(\hat{\boldsymbol{\theta}}_{\text{op}}, \boldsymbol{\theta}_{\text{gt}}) + \lambda_7 \mathcal{L}_{\text{MSE}}(\hat{\boldsymbol{\beta}}, \boldsymbol{\beta}_{\text{gt}}) \tag{6}$$

where $\lambda_6 = 1.0$ and $\lambda_7 = 0.5$ balance pose and shape parameter accuracy. This two-stage design, analytical initialization followed by learned refinement, strikes an optimal balance between efficiency and accuracy. It serves as the final validation that our learned *Sparkle* representation is not merely an intermediate output, but a powerful and efficient code for human pose and shape.

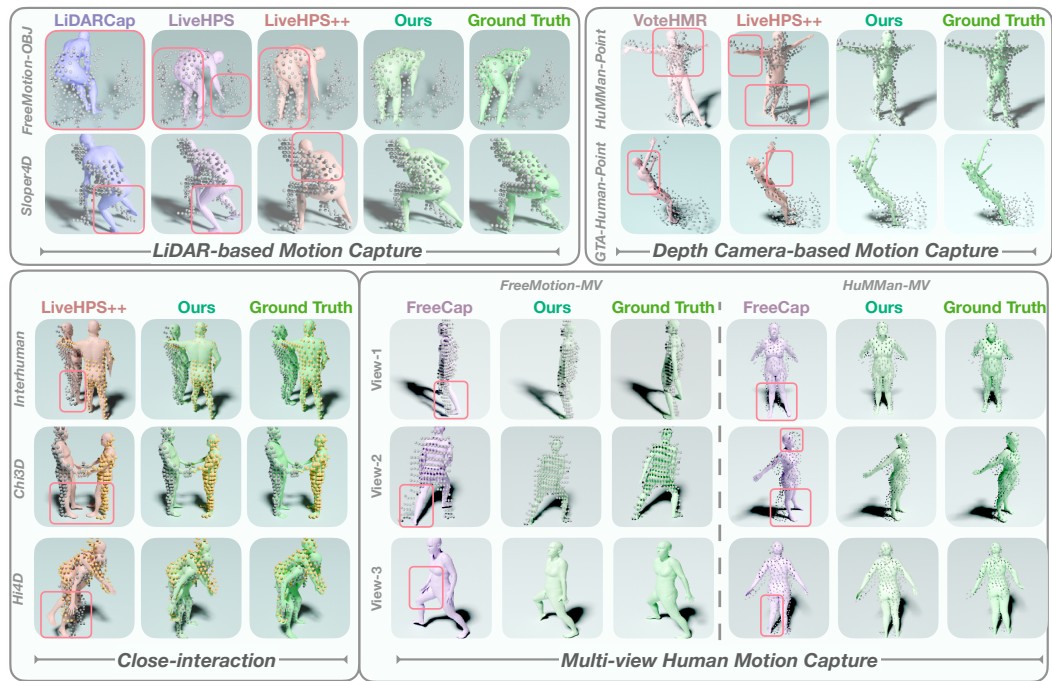

Figure 3: Qualitative comparisons. Due to the natural spatial information of point clouds, we simultaneously displayed both point clouds and global human mesh to reflect the accuracy. For multi-view MoCap, we present point clouds from three perspectives and unique fusion results from multiple perspectives. The red rounded rectangle indicates where other methods did not work correctly.

## 4 EXPERIMENT

**Implementation Details** are elaborated in the appendix.

**Datasets** 11 challenging benchmarks in total: **I). 6 datasets captured with LiDAR and depth cameras involving severe occlusion and noise**: FreeMotion(Ren et al., 2024b), FreeMotion-obj(Ren et al., 2024a), Sloper4D(Dai et al., 2023), NoiseMotion(Ren et al., 2024a), GTA-Human-Point (Cai et al., 2024), and HuMMan-Point (Cai et al., 2022), **II). 3 multi-person datasets with complex close interactions**: InterHuman(Liang et al., 2024), Chi3D (Fieraru et al., 2020) and Hi4D (Yin et al., 2023) where point cloud data is unavailable, we generate point clouds following the LIP (Ren et al., 2023). **III). 2 multi-view datasets**: FreeMotion-MV (Ren et al., 2024b) and HuMMan-MV (Cai et al., 2022).

**Baselines**: 5 point cloud-based methods, including LiDARCap (Li et al., 2022), LiveHPS (Ren et al., 2024b), VoteHMR (Liu et al., 2021), PointHPS (Cai et al., 2023) and LiveHPS++ (Ren et al., 2024a), and 1 multi-view MoCap method, FreeCap (Xue et al., 2025) Under multi-view setting.

**Evaluation Metrics** We adopt widely used metrics in motion capture to ensure a comprehensive assessment of human motion accuracy, including local and global joint/vertex errors(J/V Err(L/G))($mm \downarrow$) and angle errors(Ang Err)($degree \downarrow$).

| Method | FreeMotion (Ren et al., 2024b) | | | Sloper4D (Dai et al., 2023) | | | FreeMotion-OBJ (Ren et al., 2024a) | | | NoiseMotion (Ren et al., 2024a) | | |
|---|---|---|---|---|---|---|---|---|---|---|---|---|
| | J/V Err(L) | J/V Err(G) | Ang Err | J/V Err(L) | J/V Err(G) | Ang Err | J/V Err(L) | J/V Err(G) | Ang Err | J/V Err(L) | J/V Err(G) | Ang Err |
| LiDARCap | 86.4/104.3 | 180.4/188.6 | 15.51 | 71.6/84.2 | 138.7/147.8 | 13.72 | 84.1/100.6 | 181.8/189.3 | 16.61 | 52.6/64.7 | 400.7/402.6 | 10.87 |
| LiveHPS | 74.7/90.8 | 130.4/141.1 | 16.96 | 53.4/63.2 | 88.4/95.9 | 13.08 | 70.7/88.4 | 146.8/158.0 | 17.81 | 48.4/60.4 | 74.7/83.8 | 12.19 |
| LiveHPS++ | 61.9/75.3 | 112.1/120.4 | 15.40 | 42.7/50.6 | 77.0/81.7 | 11.92 | 58.1/72.5 | 128.60/136.94 | 15.85 | 34.0/42.8 | 58.5/64.5 | 10.63 |
| **Ours** | **59.0/73.2** | **105.1/113.9** | **9.66** | **41.4/50.5** | **70.9/77.1** | **10.48** | **50.8/62.7** | **104.1/110.5** | **8.49** | **27.8/36.3** | **38.8/45.8** | **7.57** |

Table 1: Evaluations on general scenarios with noisy and occlusion in 4 datasets. Notice our substantial improvement on noisy datasets, FreeMotion-OBJ and NoiseMotion.

## 4.1 Comparisons and Discussions

We evaluate **SparkleMotion** on the following four key capabilities:

**Robustness to Noise and Occlusion in General Scenarios.** are conducted on 2 noisy datasets, FreeMotion-obj and NoiseMotion, where point clouds contain substantial sensor noise and occlusions, as well as two normal datasets FreeMotion and Sloper4D. As shown in Table 1, our method significantly outperforms existing approaches, particularly in global joint and vertex errors (J/V Err(G)), demonstrating its ability to handle incomplete or noisy inputs. Please refer to the LiDAR-based Motion Capture of Figure 3 for a visual comparison.

**Effectiveness under Close-Interactions Scenarios.** Due to the dense proximity between individuals, occlusion and noises are more significant in such cases. In Table 2, we compare with LiveHPS++ on 3 close-interaction datasets. Our approach, significantly outperforms LiveHPS++ in challenging, occluded scenarios. By inte-

| Dataset | LiveHPS++ (Ren et al., 2024b) | | | Ours | | |
|---|---|---|---|---|---|---|
| | J/V Err(L) | J/V Err(G) | Ang Err | J/V Err(L) | J/V Err(G) | Ang Err |
| Interhuman | 41.7/62.8 | 55.0/73.8 | 18.47 | **30.4/39.9** | **40.4/48.4** | **6.75** |
| Chi3D | 39.8/62.8 | 55.0/74.4 | 23.28 | **33.7/46.4** | **47.3/57.8** | **12.09** |
| Hi4D | 47.5/71.9 | 67.9/88.8 | 25.29 | **38.6/52.4** | **54.5/66.2** | **13.11** |

Table 2: Evaluation on close-interaction datasets.

grating skeletal joints with surface keypoints, *sparkle* preserves fine motion details while ensuring structural consistency, enabling robust motion estimation. A visual comparison is shown in Figure 3.

**Cross-sensor generalization on heterogeneous point distributions** is evaluated on datasets GTA-Human-Point and HuMMan-Point, which contain data with diverse point patterns, captured with multiple point cloud-based sensors, including Ouster-1-128beam, Ouster-0-64beam, and Kinect. As shown in Table 1 and Table 3, our method consistently achieves superior performance. LiveHPS++ struggles with domain shifts between sensor types, but our approach over-

| Method | GTA-Human-Point (Cai et al., 2024) | | | HuMMan-Point (Cai et al., 2022) | | |
|---|---|---|---|---|---|---|
| | J/V Err(L) | J/V Err(G) | Ang Err | J/V Err(L) | J/V Err(G) | Ang Err |
| VoteHMR | 95.7/114.4 | -/- | - | 81.1/97.6 | -/- | - |
| PointHPS | 92.9/107.0 | -/- | - | 70.1/81.9 | -/- | - |
| LiveHPS++ | 77.5/90.4 | 127.7/137.3 | 13.17 | 71.4/92.4 | 95.6/113.1 | 21.47 |
| **Ours** | **69.1/82.2** | **122.7/131.9** | **11.52** | **63.5/76.4** | **87.5/97.6** | **12.60** |

Table 3: Evaluation on cross-sensor generalization.

comes this limitation by capturing both global motion and fine-grained details, improving robustness across diverse point patterns, refer to Figure 3.

**Adaptability to Multi-view Setups.** Our method is designed to seamlessly handle multi-view setups. In Table 4, we show our superiority on multi-view datasets FreeMotion-MV and HuMMan-MV. Unlike previous methods that rely on explicit multi-view fusion, our approach leverages

| Datasets | FreeCap (Xue et al., 2025) | | | Ours | | |
|---|---|---|---|---|---|---|
| | J/V Err(L) | J/V Err(G) | Ang Err | J/V Err(L) | J/V Err(G) | Ang Err |
| FreeMotion | 68.78/85.74 | 91.04/102.38 | 12.24 | **55.1/68.2** | **90.2/98.7** | **9.61** |
| HuMMan-Point | 72.77/87.60 | 88.07/98.06 | 13.12 | **61.8/74.3** | **83.3/93.1** | **12.14** |

Table 4: Evaluation on multi-view datasets.

*sparkle*-based point cloud segmentation results, which predict the confidence levels of keypoints. This confidence allows us to dynamically select the most reliable combination of *sparkle* representations across multiple viewpoints, significantly improving accuracy, as illustrated in Fig. 3.

## 4.2 Ablation Studies

| Datasets | Network Structure | | | Point-alighed Skeleton Tracker (PST) | | Skeleton-guided Anchor Estimator (SAE) | | Sparkle-based SMPL Solver (SSS) | | Ours |
|---|---|---|---|---|---|---|---|---|---|---|
| | w/o PST | w/o SAE | w/o SSS | w/o Offset | w/o OP | w/o Initialization | w/o OP | w/o Initialization | w/o OP | |
| FreeMotion | 149.2/159.6 | 116.0/125.3 | 112.6/122.4 | 107.2/116.7 | 106.3/115.4 | 108.3/118.4 | 107.8/117.0 | 108.8/117.9 | 106.3/119.6 | **105.1/113.9** |
| HuMMan-Point | 112.3/123.9 | 95.3/108.6 | 91.9/101.6 | 89.7/101.3 | 91.2/102.3 | 89.4/99.1 | 90.4/100.2 | 91.9/102.0 | 88.0/101.1 | **87.5/97.6** |
| Interhuman | 58.4/66.1 | 55.3/64.9 | 56.7/67.2 | 43.2/52.0 | 42.2/50.3 | 44.1/53.3 | 42.6/49.8 | 43.9/51.6 | 46.4/62.9 | **40.4/48.4** |
| FreeMotion-MV | 135.4/142.8 | 99.7/107.5 | 95.4/104.3 | 92.4/100.5 | 92.0/99.8 | 93.6/102.0 | 92.6/101.2 | 93.9/102.7 | 92.1/100.7 | **90.2/98.7** |

Table 5: Ablation studies for our network modules across multiple datasets.

We conduct extensive ablation studies on **5** datasets FreeMotion, FreeMotion-MV, GTA-Human-Point, and InterHuman to validate the effectiveness of each component in *sparklemotion* (Table 5).

**Network Structure.** Without PST, skeletal estimation degrades with increased global errors due to the lack of explicit point-joint correspondence modeling; without SAE, pose accuracy suffers as the model fails to leverage surface geometry for resolving rotational ambiguities; without SSS, SMPL

parameter regression becomes less accurate without exploiting the kinematic-geometric factorization of the *Sparkle* representation.

**Component-level Analysis. PST:** Replacing the offset-based refinement proves with direct prediction (w/o offset) increases joint error, as the residual learning paradigm simplifies the complex regression task. Removing the optimization (w/o op) further degrades global alignment, emphasizing the importance of iterative joint-point cloud registration. **SAE:** Skipping linear initialization (w/o initialize) leads to unstable convergence, while omitting the non-linear refinement (w/o op) cause an increase in anchor error, demonstrating that the cross-attention based correction effectively fuses structural priors with geometric evidence. **SSS:** Removing the geometric initialization (w/o init) causes the network to converge to suboptimal poses, while skipping the refinement (w/o op) leaves errors from the analytical solution uncorrected, particularly for complex shapes.

| Datasets | PCA Selection | | | Random Selection | | | | Manual Selection | | | Ours |
|---|---|---|---|---|---|---|---|---|---|---|---|
| | 16 Anchors | 64 Anchors | 96 Anchors | 16 Anchors | 32 Anchors | 64 Anchors | 96 Anchors | 41 Anchors | 50 Anchors | 60 Anchors | |
| FreeMotion | 112.7/121.0 | 107.4/115.3 | 110.1/118.9 | 110.8/118.7 | 117.2/126.3 | 116.4/125.0 | 111.2/118.7 | **104.8**/114.1 | 105.8/114.7 | 107.3/114.9 | 105.1/**113.9** |
| HuMMan-Point | 93.1/104.2 | 94.2/105.1 | 95.7/106.3 | 91.6/101.6 | 94.6/103.8 | 90.2/99.7 | 91.1/102.4 | 87.1/97.0 | 86.5/97.2 | **85.3/96.4** | 87.5/97.6 |
| Interhuman | 50.2/58.4 | 47.7/55.1 | 49.6/57.1 | 51.3/58.6 | 52.2/60.0 | 49.3/56.4 | 51.7/58.6 | 40.8/48.6 | 41.2/49.1 | 48.0/55.9 | **40.4/48.4** |
| FreeMotion-MV | 97.4/106.3 | 93.2/102.7 | 95.3/104.1 | 95.2/104.1 | 102.6/113.1 | 102.4/111.7 | 96.9/105.2 | **89.3/97.2** | 90.1/98.0 | 92.4/100.1 | 90.2/98.7 |

Table 6: Ablation studies for our network in different selections and numbers of surface anchors.

**Analysis of Anchor Design.** A core component of our Sparkle representation is the set of surface anchors. To validate our design choices, we conduct a comprehensive ablation study on the number and selection strategy of these anchors. We evaluate three strategies: **PCA-based selection**, which provides a data-driven, compact representation of surface geometry; **Random selection** from SMPL vertices; and **Manual selection** based on the CMU (Carnegie Mellon University) marker set, representing an anatomically-informed prior. For each strategy, we vary the number of anchors and report the performance on the FreeMotion dataset, with results summarized in Table 6. Our findings conclusively demonstrate the superiority of the PCA-based strategy with 32 anchors. The PCA-16 configuration suffers from insufficient coverage of surface details, leading to a noticeable performance drop. Conversely, configurations with more anchors (PCA-64 and PCA-96) also exhibit degraded performance. Predicting an excessive number of anchors from noisy, sparse point clouds makes the regression task more challenging and prone to overfitting, as errors in estimating individual anchors accumulate. The random selection strategy yields unstable and suboptimal results. Its performance fluctuates irregularly with the number of anchors, as the random distribution may over-cover certain body regions while neglecting others. For instance, the Random-64 configuration performed worse than Random-16 due to its uneven coverage. While the manual selection strategy provides robust performance due to its comprehensive anatomical coverage (even slightly outperforming our default setup in some cases), the improvement is marginal. More importantly, it requires manual selection and lacks the generalizability and automation of our data-driven PCA approach. Therefore, the PCA-32 configuration strikes the optimal balance between representational capacity, robustness to noise, and computational efficiency. These experiments robustly validate that our anchor module, by augmenting skeletal joints with a well-chosen set of surface anchors, is instrumental in constructing a more expressive and robust representation for human motion capture.

## 5 CONCLUSION

In conclusion, *Sparkle* novelly unifies internal kinematics and surface geometry to advance point cloud-based motion capture. *SparkleMotion* leverages this representation to achieve stable, accurate motion estimation by dynamically balancing structural consistency and fine-grained motion details. Through extensive experiments across multiple datasets, we demonstrate that *SparkleMotion* achieves state-of-the-art performance in terms of robustness to noise and occlusion, effectiveness under close-interaction scenarios, generalization across diverse point cloud patterns, and adaptability to different capture setups. Finally, the real-time deployment of *SparkleMotion* achieves high frame rates, making it well-suited for practical, large-scale motion capture in diverse real-world settings. **The real-time demos in the supplementary material videos.**

ETHICS STATEMENT

This research focuses on motion capture from point clouds, which, by their nature, contain less identifiable information (e.g., faces, skin texture) compared to RGB video, offering a inherent privacy benefit. We envision positive applications in virtual reality, human-robot interaction, and sports analysis.

Nonetheless, we acknowledge the potential for any perception technology to be misused. While the privacy risk is lower, detailed human motion data could potentially be used for malicious surveillance or creating deepfakes. To mitigate these risks, the code and models we release are intended solely for research purposes, and the license will explicitly prohibit use for privacy infringement, creating misleading content, or any harmful applications. We encourage the community to consider principles of fairness, accountability, and transparency when building upon this work and to establish appropriate governance frameworks to prevent misuse.

REPRODUCIBILITY STATEMENT

To ensure the reproducibility of our work, we commit to the following:

Code and Models: We will open-source the complete codebase for training and inference, along with the pre-trained model weights for the best-performing checkpoint on all benchmarks after paper acceptance.

**Data**: Our experiments are conducted entirely on publicly available datasets (e.g., Freemotion, HuMMan, SLOPER4D). All sources are cited in the paper. For datasets involving point cloud generation (e.g., InterHuman), we have detailed the data pre-processing pipeline following public methods.

**Implementation Details**: The methodology section 3 and the supplementary material A provide comprehensive descriptions of the network architectures, loss functions, and training hyperparameters (e.g., learning rate, batch size, optimizer). All critical parameters are explicitly listed.

**Dependencies**: The code repository will include a detailed requirements.txt file and/or a Docker image to specify the required software libraries (e.g., PyTorch, CUDA) and their versions, ensuring a consistent computational environment.

We are confident that these measures are sufficient for other researchers to replicate our results.

ACKNOWLEDGEMENTS

This work was supported by MoE Key Laboratory of Intelligent Perception and Human-Machine Collaboration (KLIP-HuMaCo), Shanghai Frontiers Science Center of Human-centered Artificial Intelligence (ShangHAI).

This research is supported by the National Research Foundation, Singapore and Infocomm Media Development Authority under its Trust Tech Funding Initiative. Any opinions, findings and conclusions or recommendations expressed in this material are those of the author(s) and do not reflect the views of National Research Foundation, Singapore and Infocomm Media Development Authority.

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

APPENDIX

## A  IMPLEMENTATION AND NETWORK DETAILS

### A.1  IMPLEMENTATION DETAILS

Our method is implemented using PyTorch 1.8.1 and CUDA 11.1. We train the model for 200 epochs with a batch size of 32 and a sequence length of 32, using an initial learning rate of 0.001. The training was conducted on a server with an Intel(R) Xeon(R) Gold 5318Y CPU and 4 NVIDIA A40 GPUs. For LiDAR-based datasets, we follow the settings from LiveHPS++ (Ren et al., 2024b). The model is trained using the training datasets listed in the table and tested on their respective test sets. For Depth camera-based datasets, we adopt the training setup from PointHPS (Cai et al., 2023), training and testing each benchmark separately.

### A.2  DETAILED ARCHITECTURE OF POINT-ALIGNED SKELETON TRACKER

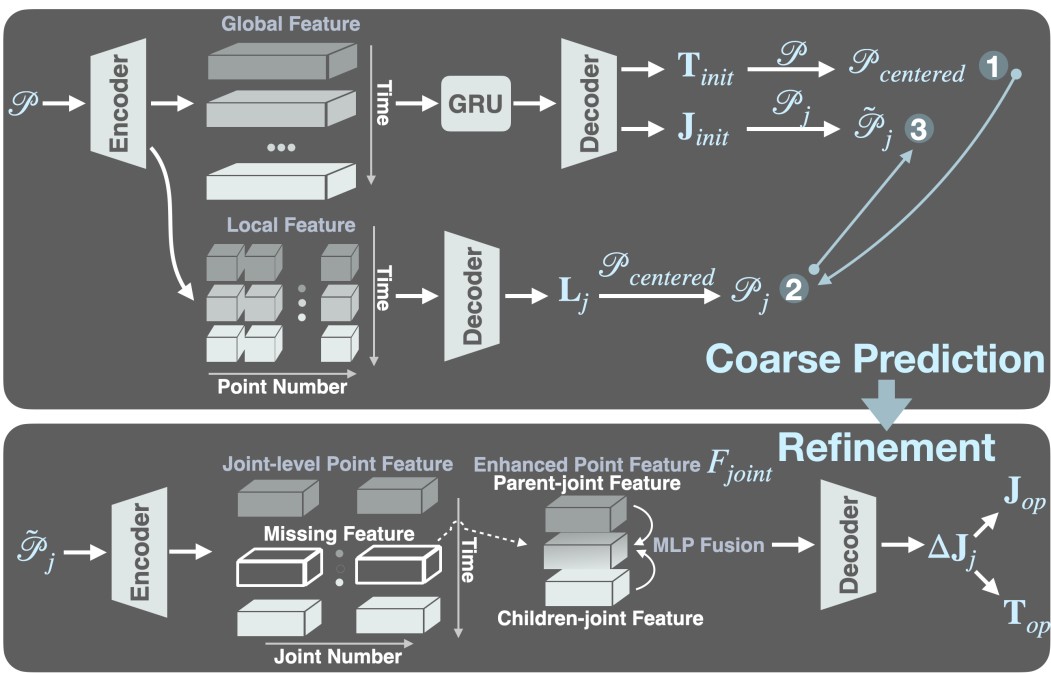

Figure 4: The details of Point-aligned Skeleton Tracker.

The Point-aligned Skeleton Tracker (PST) is implemented as a dual-branch network that jointly estimates initial skeletal joints $\mathbf{J}_{\text{init}}$, global translation $\mathbf{T}_{\text{init}}$, and point-wise semantic labels $\mathbf{L}_j$ from the input point cloud sequence. Its architecture comprises two main components as shown in Fig 4:

**1. Coarse Prediction Network:** This branch processes the spatio-temporal point cloud sequence. Given an input point cloud $\mathcal{P}$ of shape $(B, T, 3, N)$ (Batch, Time, Channels, Points), a *PointNet* backbone first extracts per-frame features. A bidirectional GRU with a hidden size of 512 (yielding a 1024-dimensional output due to bidirectionality) models temporal dependencies. The network also ingests a trajectory signal, which is encoded by MLPs ($3 \rightarrow 64 \rightarrow 128$) and concatenated with the point cloud features. The branch outputs:

- Initial joints $\mathbf{J}_{\text{init}}$: Regressed via an MLP ($1024 \rightarrow 512 \rightarrow 256 \rightarrow 24 \times 3$).

- Initial translation $\mathbf{T}_{\text{init}}$ (optional): Regressed via a parallel MLP ($1024 \rightarrow 512 \rightarrow 256 \rightarrow 3$).

- Point-wise part labels $\mathbf{L}_j$: A segmentation head, consisting of 1D convolutional layers $(2048 \rightarrow 1024 \rightarrow 512 \rightarrow 256 \rightarrow k)$, predicts a label for each of the $N$ points, where $k = 36$ is the number of semantic parts (including background).

**2. Refinement Network:** This branch refines the initial estimates using local geometric context. It takes as input the raw point cloud $\mathcal{P}$, the initial skeleton $\mathbf{J}_{\text{init}}$, and the predicted part labels $\mathbf{L}_j$. For each point, it constructs a 6D feature by concatenating its spatial coordinates with the coordinates of its associated joint (as indicated by $\mathbf{L}_j$). A *PointNet* processes this enriched input to extract local features. A key operation is a *label-wise max-pooling*: for each of the 24 joints, features from all points belonging to that joint part are pooled to form a robust, joint-centric feature descriptor. These descriptors are then decoded by an MLP $(1024 \rightarrow 512 \rightarrow 256 \rightarrow 3)$ to predict the residual joint offsets $\Delta\mathbf{J}$, yielding the final refined joints $\mathbf{J}_{\text{op}} = \mathbf{J}_{\text{init}} + \Delta\mathbf{J}$ and translations $\mathbf{T}_{\text{op}} = \mathbf{T}_{\text{init}} + \Delta\mathbf{J_0}$.

The PST module effectively couples a global spatio-temporal model for coarse estimation with a local geometry-aware network for fine-grained refinement, establishing the explicit point-joint correspondences central to our method.

### A.3 Detailed Architecture of Skeleton-guided Anchor Estimator

#### A.3.1 Structural Initialization via Vertex-Joint Matching

Prior to training, we precompute a linear mapping between skeletal joints and surface anchors to provide a strong structural initialization. This process begins with **Vertex-Joint Matching** on the SMPL mesh, where each vertex is associated with its nearest skeletal joint, establishing a **Joint-level Segmentation** of the mesh surface. From the full set of mesh vertices, we select a canonical set of 32 anchors $\mathbf{A}_{\text{gt}}$ via PCA to maximize their representational power. A linear mapping matrix $\mathbf{M}_{\text{J2A}} \in \mathbb{R}^{24 \times 32}$ is then computed in a least-squares manner from ground-truth data:

$$\mathbf{M}_{\text{J2A}} = (\mathbf{A}_{\text{gt}}^\top \mathbf{A}_{\text{gt}})^{-1} \mathbf{A}_{\text{gt}}^\top \mathbf{J}_{\text{gt}}.$$

During inference, this matrix provides a geometry-agnostic prior, yielding an initial anchor estimate based solely on the predicted skeletal pose: $\mathbf{A}_{\text{init}} = \mathbf{J}_{\text{op}}\mathbf{M}_{\text{J2A}}$.

#### A.3.2 Geometry-Aware Refinement Network

The initial estimate $\mathbf{A}_{\text{init}}$ is refined by a network that learns a non-linear correction using geometric evidence from the predicted joint $\mathbf{J}_{op}$. The core of this network is a **Transformer-based decoder** that performs cross-attention between anchor queries and geometric features.

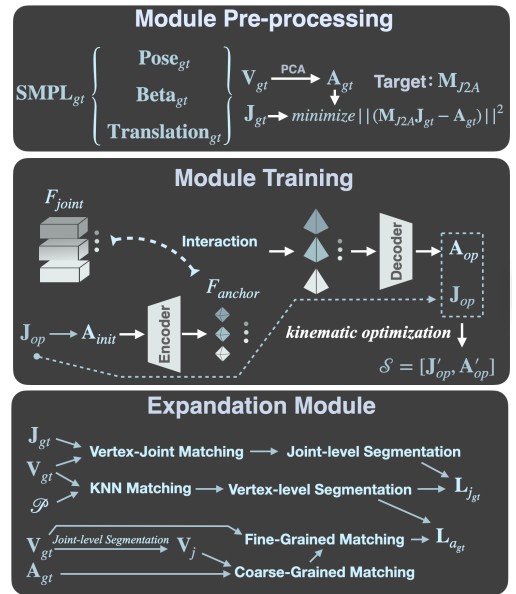

Figure 5: The details of Skeleton-guided Anchor Estimator.

**Feature Encoding:** A *MLP* backbone processes the predicted joints to extract local geometric features. Simultaneously, the initial anchors $\mathbf{A}_{\text{init}}$ are encoded into a 1024-dimensional feature space via an MLP $(3 \rightarrow 256 \rightarrow 512 \rightarrow 1024)$.

**Cross-Attention Refinement:** The refinement is formulated as a sequence-to-sequence problem. The encoded initial anchors serve as the *query* sequence (32 anchors, dimension 1024). The joint features from the MLP serve as the *key* and *value* sequence. A 2-layer Transformer decoder with 8 attention heads performs cross-attention, allowing each anchor query to attend to the most relevant joint-level geometric context. The attended features are then decoded by an MLP $(1024 \rightarrow 512 \rightarrow 256 \rightarrow 3)$ to predict the optimized anchors $\mathbf{A}_{\text{op}}$

After we get the predicted joints and anchors, we use the Kinematic-aware Optimizer from LiveHPS++ Ren et al. (2024a) to optimize the *Sparkle* from the raw point cloud data and fully utilize the temporal information.

### A.3.3    Anchor-level Segmentation for Expandsion

The module also predicts an **anchor-level segmentation** of the point cloud, which is crucial for downstream tasks like multi-view fusion. The generation of the ground-truth segmentation labels $\mathbf{L}_{A_{gt}}$ for training is a multi-stage process that ensures precise correspondence between points and anchors, as illustrated in Figure 5.

1. **Vertex-Joint Matching on SMPL Mesh:** The process begins on the canonical SMPL mesh. We compute the Euclidean distance from each mesh vertex to all skeletal joints. Each vertex is then assigned to its closest joint, establishing a fundamental **Joint-level Segmentation** at the mesh level. This defines a surjective mapping from the set of vertices to the set of joints.

2. **Point Cloud to Vertex Correspondence (Coarse-Grained Matching):** To propagate the mesh-based segmentation to an unstructured point cloud $\mathcal{P}_t$, we establish correspondence between points and mesh vertices. This is achieved via a **KNN Matching** algorithm, where each point in $\mathcal{P}_t$ is matched to its $k$ nearest neighbors in the set of SMPL mesh vertices. The joint label for a point is determined by a majority vote among the labels of its $k$ nearest vertices. This step effectively projects the mesh's **Joint-level Segmentation** onto the point cloud, yielding the coarse-grained point-wise labels $\mathbf{L}_{j_{gt}}$.

3. **Anchor to Vertex Assignment:** The 32 surface anchors $\mathbf{A}_{gt}$ are derived from the mesh vertices via PCA. Crucially, each anchor is assigned to a specific joint based on the pre-computed **Vertex-Joint Matching**. Specifically, an anchor is assigned the joint label of the mesh vertex from which it was predominantly derived. This creates a well-defined, many-to-one mapping from anchors to joints.

4. **Point to Anchor Correspondence (Fine-Grained Matching):** The final **Anchor-level Segmentation** $\mathbf{L}_{A_{gt}}$ is obtained by performing a **fine-grained matching** within the confines of the coarse segmentation. For each point in $\mathcal{P}_t$, its anchor label is determined by finding its nearest anchor *from the subset of anchors that share the same joint label as the point itself* (as determined by $\mathbf{L}_{j_{gt}}$). This **joint-localized KNN** is critical, as it prevents erroneous matches that could occur when a point is physically closer to an anchor belonging to a neighboring joint (e.g., a point on the wrist mistakenly matched to a leg anchor).

This hierarchical labeling strategy ensures that the anchor-level segmentation is both geometrically precise and anatomically consistent. The network learns to predict this segmentation through a parallel segmentation head, composed of 1D convolutional layers ($2048 \rightarrow 1024 \rightarrow 512 \rightarrow 256 \rightarrow k$). The quality of the predicted segmentation $\mathbf{L}_a$ implicitly defines a confidence measure for each predicted anchor, enabling downstream modules (e.g., the multi-view fusion logic) to dynamically weight the reliability of the *Sparkle* representation.

### A.4    Rotation Decomposition in Sparkle-based SMPL Solver

We detail the initialization of SMPL pose in Sparkel-based SMPL Solver. The rotation $R \in SO(3)$ can be decomposed into a twist rotation $R_{tw}$ and a swing rotation $R_{sw}$. Given the start template body-part joint vector $\vec{J}_{\text{tem}}$, marker vector $\vec{A}_{\text{tem}}$, target joint vector $\vec{J}'_{\text{op}}$, and target marker vector $\vec{A}'_{\text{op}}$, the solution process for $R$ can be formulated as:

$$\begin{aligned}
R &= \mathcal{D}(\vec{J}_{\text{tem}}, \vec{A}_{\text{tem}}, \vec{J}'_{\text{op}}, \vec{A}'_{\text{op}}) \\
&= \mathcal{D}^{\text{sw}}(\vec{J}_{\text{tem}}, \vec{J}'_{\text{op}}) \cdot \mathcal{D}^{\text{tw}}(\vec{J}'_{\text{op}}, \vec{A}_{\text{tem}}, \vec{A}'_{\text{op}}) \\
&= R^{\text{sw}} \cdot R^{\text{tw}}
\end{aligned}$$

where $\mathcal{D}^{\text{sw}}$ and $\mathcal{D}^{\text{tw}}$ represent the closed-form solutions for the swing rotation and twist rotation, respectively. The swing rotation has an axis $\vec{n}_{\text{sw}}$ that is perpendicular to both $\vec{J}_{\text{tem}}$ and $\vec{J}'_{\text{op}}$. It can be

formulated as:

$$\vec{n}_{\text{sw}} = \frac{\vec{J}_{\text{tem}} \times \vec{J}'_{\text{op}}}{\|\vec{J}_{\text{tem}} \times \vec{J}'_{\text{op}}\|}$$

The swing angle $\alpha_{\text{sw}}$ satisfies:

$$\cos \alpha_{\text{sw}} = \frac{\vec{J}_{\text{tem}} \cdot \vec{J}'_{\text{op}}}{\|\vec{J}_{\text{tem}}\| \|\vec{J}_{\text{op}'}\|},$$

$$\sin \alpha_{\text{sw}} = \frac{\|\vec{J}_{\text{tem}} \times \vec{J}'_{\text{op}}\|}{\|\vec{J}_{\text{tem}}\| \|\vec{J}'_{\text{op}}\|}$$

Thus, the closed-form solution for the swing rotation $R^{\text{sw}}$ can be derived using the Rodrigues formula:

$$R^{\text{sw}} = \mathcal{D}^{\text{sw}}(\vec{J}_{\text{tem}}, \vec{J}'_{\text{op}})$$
$$= \mathcal{I} + \sin \alpha_{\text{sw}}[\vec{n}_{\text{sw}}]_{\times} + (1 - \cos \alpha_{\text{sw}})[\vec{n}_{\text{sw}}]^2_{\times}$$

where $[\vec{n}_{\text{sw}}]_{\times}$ is the skew-symmetric matrix of $\vec{n}_{\text{sw}}$, and $\mathcal{I}$ is the $3 \times 3$ identity matrix. The twist rotation has an axis $\vec{n}_{\text{tw}}$ defined as:

$$\vec{n}_{\text{tw}} = \frac{\vec{J}'_{\text{op}}}{\|\vec{J}'_{\text{op}}\|}$$

The twist angle $\alpha_{\text{tw}}$ satisfies:

$$\cos \alpha_{\text{tw}} = \frac{\vec{A}_{\text{tem}}^{\text{proj}} \cdot \vec{A}_{\text{op}}'^{\text{proj}}}{\|\vec{A}_{\text{tem}}^{\text{proj}}\| \|\vec{A}_{\text{op}}'^{\text{proj}}\|},$$

$$\sin \alpha_{\text{tw}} = \frac{\|\vec{A}_{\text{tem}}^{\text{proj}} \times \vec{A}_{\text{op}}'^{\text{proj}}\|}{\|\vec{A}_{\text{tem}}^{\text{proj}}\| \|\vec{A}_{\text{op}}'^{\text{proj}}\|}$$

where $\vec{A}_{\text{tem}}^{\text{proj}}$ and $\vec{A}_{\text{op}}'^{\text{proj}}$ are projections onto the plane perpendicular to the axis $\vec{n}_{\text{tw}}$:

$$\vec{A}_{\text{tem}}^{\text{proj}} = \vec{A}_{\text{tem}} - \left(\frac{\vec{A}_{\text{tem}} \cdot \vec{n}_{\text{tw}}}{\|\vec{n}_{\text{tw}}\|}\right) \vec{n}_{\text{tw}},$$

$$\vec{A}_{\text{op}}'^{\text{proj}} = \vec{A}'_{\text{op}} - \left(\frac{\vec{A}'_{\text{op}} \cdot \vec{n}_{\text{tw}}}{\|\vec{n}_{\text{tw}}\|}\right) \vec{n}_{\text{tw}}$$

Thus, the closed-form solution for the twist rotation $R^{\text{tw}}$ can be derived using the Rodrigues formula:

$$R^{\text{tw}} = \mathcal{D}^{\text{tw}}(\vec{J}'_{\text{op}}, \vec{A}_{\text{tem}}, \vec{A}'_{\text{op}})$$
$$= \mathcal{I} + \sin \alpha_{\text{tw}}[\vec{n}_{\text{tw}}]_{\times} + (1 - \cos \alpha_{\text{tw}})[\vec{n}_{\text{tw}}]^2_{\times}$$

where $[\vec{n}_{\text{tw}}]_{\times}$ is the skew-symmetric matrix of $\vec{n}_{\text{tw}}$.

# B  DETAILS OF MULTI-VIEW MOCAP BY SPARKLEMOTION

Our method also can handle **arbitrary-viewpoint** point clouds as input and robustly fuse information from multiple perspectives. Given a set of multi-view point clouds $\{\mathcal{P}_i\}_{i=1}^N$ captured from different viewpoints, we aim to obtain a globally consistent human motion representation while maintaining robustness to occlusions, sensor noise, and varying point distributions.

**View-Independent Sparkle Prediction** For each individual view $\mathcal{P}_i$, we utilize our trained **SparkleMotion** model to predict the segmentation result, the corresponding *Sparkle* representation and global human rotation $\mathcal{S}_i$:

$$\mathcal{S}_i = \mathcal{F}_{\text{SparkleMotion}}(\mathcal{P}_i), \tag{7}$$

where $\mathcal{S}_i$ consists of:

- **Segmented point cloud assignments** from the input.

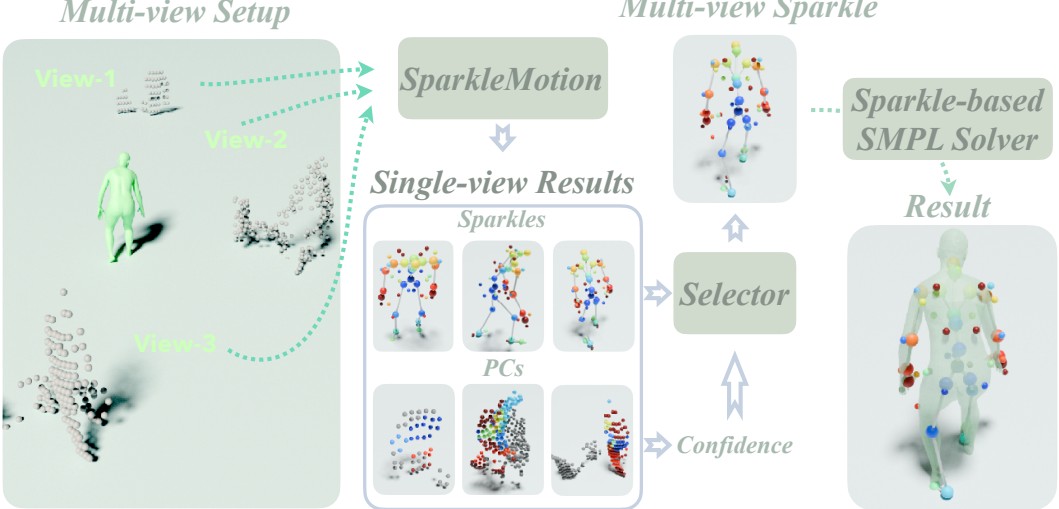

Figure 6: The pipeline of the multi-view SparkleMotion. Our method can take point clouds from any perspective as input to obtain optimized results, demonstrating the powerful scalability of *Sparkle*.

- **Root-relative *sparkle*** capturing local motion structure.
- **Global rotation estimation** to align multiple views.

Each predicted *sparkle* is assigned a confidence score $\mathbf{w}_i$ based on segmentation quality, reflecting its reliability:

$$\mathbf{w}_i = \mathcal{G}(\mathcal{P}_i, \mathcal{S}_i), \tag{8}$$

where $\mathcal{G}$ represents our confidence estimation function, which considers point density, occlusions, and consistency in spatial feature aggregation.

**Multi-View alignment via global rotation estimation** Before fusing the information across views, we first **calibrate** each predicted **global rotation** $\mathbf{R}_i$. Each viewpoint provides an independent estimate of global orientation, and we refine it by minimizing the misalignment across different views:

$$\mathbf{R}^* = \arg\min_{\mathbf{R}} \sum_{i=1}^{N} \|\mathbf{R} - \mathbf{R}_i\|^2, \tag{9}$$

where $\mathbf{R}^*$ is the optimized global rotation used for aligning all views.

**Confidence-Guided Multi-View Fusion** Once the multi-view predictions $\{\mathcal{S}_i, \mathbf{w}_i\}_{i=1}^{N}$ are obtained, we perform confidence-based selection to **filter out unreliable keypoints** and obtain a refined *sparkle*:

$$\mathcal{S}^* = \sum_{i=1}^{N} \mathbf{w}_i \cdot \mathcal{S}_i. \tag{10}$$

This ensures that **keypoints predicted with higher confidence** have greater influence, while uncertain keypoints from occluded or noisy views are suppressed.

**Sparkle-Based SMPL Parameter Estimation** Finally, the optimized *sparkle* $\mathcal{S}^*$ is fed into our **Sparkle-based SMPL Solver** to predict human body parameters:

$$\Theta^* = \mathcal{H}(\mathcal{S}^*), \tag{11}$$

where $\Theta^*$ includes body shape $\beta$, pose parameters $\theta$, and global translation $\mathbf{t}$. Our solver ensures that the final motion estimation maintains structural consistency while fully utilizing the information across views.

Multi-view motion capture is essential for achieving robust and accurate human motion estimation, especially in real-world scenarios with occlusions and varying viewpoints. Traditional methods often rely on explicit camera calibration and heuristic fusion strategies, limiting their flexibility and

adaptability. Our proposed framework, **Sparklemotion**, introduces a segmentation-driven confidence mechanism and a unified motion representation, allowing seamless integration of multiple views without requiring predefined camera setups. By dynamically filtering unreliable keypoints and leveraging hierarchical optimization, **Sparklemotion** ensures scalable, robust, and generalizable motion capture across diverse capture conditions, from sparse LiDAR data to dense depth-camera inputs.

## C  ANALYSIS ON DISTANCE AND OCCLUSION

This section provides a quantitative analysis of the system's performance under challenging conditions, specifically increasing capture distance and varying degrees of occlusion. These scenarios directly test the stability of the geometric solver, as they often lead to sparser point clouds, noisier *Sparkle* estimates, and situations where joints or anchors may become occluded or their observations may become numerically unstable (e.g., near-colinear anchors).

### C.1  PERFORMANCE IN DIFFERENT CAPTURE DISTANCE

We first evaluate the robustness of *SparkleMotion* on the FreeMotion dataset by stratifying the test results based on the distance between the subject and the LiDAR sensor. Increased distance leads to sparser point clouds, making the estimation of both joints and anchors more challenging. The results are presented in Table 7.

| Distance Range | J/V Err(L) | J/V Err(G) | Ang Err |
|---|---|---|---|
| 5m - 10m | 52.6/65.4 | 89.9/98.3 | 9.01 |
| 10m - 20m | 61.6/76.1 | 115.4/123.6 | 9.99 |
| 20m - 30m | 63.7/79.0 | 114.2/124.0 | 10.04 |

Table 7: Performance analysis under different capture distances on the FreeMotion dataset.

The results indicate a graceful degradation in performance as distance increases. The error metrics see the most significant jump from the 5-10m to the 10-20m range, which aligns with the quadratic fall-off in point density with distance. Notably, the performance stabilizes between the 10-20m and 20-30m ranges, demonstrating the resilience of the *Sparkle* representation and the two-stage solver. The kinematic prior and the refinement network effectively prevent a complete failure even with very sparse data.

### C.2  PERFORMANCE IN DIFFERENT OCCLUSION RATIO

To directly analyze the stability under the **nearly colinear or occluded** conditions, we evaluate performance on sequences with artificially generated occlusion. We simulate different occlusion levels by randomly removing a specified percentage of points from the human point cloud. The results are summarized in Table 8.

| Occlusion Ratio | J/V Err(L) | J/V Err(G) | Ang Err |
|---|---|---|---|
| 0% | 59.0/73.2 | 105.1/113.9 | 9.66 |
| 30% | 60.0/74.4 | 107.8/116.7 | 9.81 |
| 50% | 61.7/76.5 | 110.8/119.8 | 10.07 |
| 70% | 66.7/82.6 | 118.7/128.3 | 10.92 |
| 90% | 97.4/121.2 | 163.3/179.0 | 14.18 |

Table 8: Performance analysis under different occlusion ratios. Metrics are reported on a held-out test set with simulated occlusion.

As shown in Table 8, the system maintains reasonable accuracy up to 70% occlusion. The degradation in performance is monotonic and controlled. The increase in AngErr under heavy occlusion (90%) reflects the increased ambiguity in estimating the twist component of the rotation when surface anchors are missing. This empirically validates that our two-stage design, where the refinement

network corrects for failures of the geometric initializer, provides significant robustness against the instability of the analytical solver under severe occlusion.

## C.3 DETAILED ABLATION STUDIES ON PST AND SAE MODULE DESIGNS

This section provides a more granular analysis of the design choices within the Point-aligned Skeleton Tracker (PST) and the Skeleton-guided Anchor Estimator (SAE), extending the coarse-grained ablation studies in the main paper.

### C.3.1 PST REFINEMENT STRATEGY ANALYSIS

We evaluated several refinement strategies for the PST module beyond the simple direct prediction baseline. The results on the FreeMotion dataset are summarized in Table 9.

| Refinement Strategy | J/V Err(L) | J/V Err(G) | Ang Err |
|---|---|---|---|
| Direct Prediction (w/o offset) | 68.1/83.5 | 107.2/116.7 | 10.23 |
| Iterative Refinement (2 stages) | **58.5/72.8** | **104.8/113.5** | **9.61** |
| Attention-based Refinement | 59.8/74.1 | 106.2/114.9 | 9.80 |
| **Residual Offset (Ours)** | 59.0/73.2 | 105.1/113.9 | 9.66 |

Table 9: Detailed ablation of different refinement strategies for the PST module on the FreeMotion dataset.

The residual learning paradigm proves most effective, substantially outperforming direct prediction by simplifying the regression task, while both iterative and attention-based refinements yield diminishing returns relative to their increased computational complexity.

### C.3.2 SAE INITIALIZATION AND REFINEMENT ANALYSIS

We further dissect the SAE module by evaluating different initialization and refinement strategies. Results are reported on the FreeMotion dataset in Table 10.

| Method | J/V Err(L) | J/V Err(G) |
|---|---|---|
| *A. Initialization Strategy Analysis* | | |
| Random Initialization | 95.4/115.2 | 155.7/168.3 |
| MLP-based Initialization | 65.3/80.1 | 112.4/121.0 |
| **Linear Mapping (Ours)** | **59.0/73.2** | **105.1/113.9** |
| *B. Refinement Strategy Analysis (using Linear Mapping Init)* | | |
| No Refinement (w/o op) | 66.2/81.0 | 107.8/117.0 |
| MLP Refinement | 61.5/75.8 | 106.9/115.6 |
| **Cross-Attention Refinement (Ours)** | **59.0/73.2** | **105.1/113.9** |

Table 10: Detailed ablation of initialization and refinement strategies for the SAE module on the FreeMotion dataset.

Our linear mapping initialization establishes a robust anatomical prior that significantly outperforms alternatives, while the cross-attention refinement proves essential for achieving precise, context-aware corrections beyond what standard MLP refinement can offer.

## D ANALYSIS OF DATA EFFICIENCY AND GENERALIZATION

To rigorously evaluate both the generalization capability and data efficiency of the proposed representation, we conduct a cross-dataset transfer learning experiment. A model is first pre-trained on a mixture of datasets excluding FreeMotion, and then fine-tuned on progressively

larger fractions of the FreeMotion training set before evaluation on the held-out FreeMotion test set. As summarized in Table 11, SparkleMotion demonstrates remarkable zero-shot generalization when no FreeMotion data is used (0%), significantly outperforming the baseline. Furthermore, it exhibits superior data efficiency, achieving performance comparable to the fully supervised baseline using only 50% of the target domain training data.

Theoretically, these empirical advantages stem from the structured prior embedded in the Sparkle representation, which effectively reduces the hypothesis space and enhances identifiability. By explicitly factorizing the human state into internal kinematics (skeletal joints) and external geometry (surface anchors), we introduce a strong inductive bias that disentangles pose from shape variations. This factorization mitigates the ill-posedness inherent in estimating human pose from partial or noisy point clouds, as the kinematic structure provides robust constraints that are largely invariant to sensor type or surface noise. From a learning theory perspective, this structural disentanglement simplifies the mapping function that the network must learn, thereby lowering sample complexity and enabling more efficient knowledge transfer

| T-Data | LiveHPS++ | Ours |
|--------|-----------|------|
| 0% | 84.0 | **65.2** |
| 25% | 78.4 | **63.1** |
| 50% | 68.9 | **61.5** |
| 75% | 63.2 | **59.8** |
| 100% | 61.9 | **59.0** |

Table 11: Cross-dataset generalization and data efficiency.

across domains. The consistent skeletal topology serves as a domain-invariant feature, while the surface anchors capture sensor-specific geometries without corrupting the structural estimate. This result provides strong evidence that the structural prior embedded in the Sparkle representation is not only robust to domain shifts but also drastically reduces the required amount of labeled data for achieving state-of-the-art performance, confirming its effectiveness as a powerful and efficient representation-learning principle for 3D human motion capture.

# E REAL-WORLD DEPLOYMENT AND BENCHMARKING

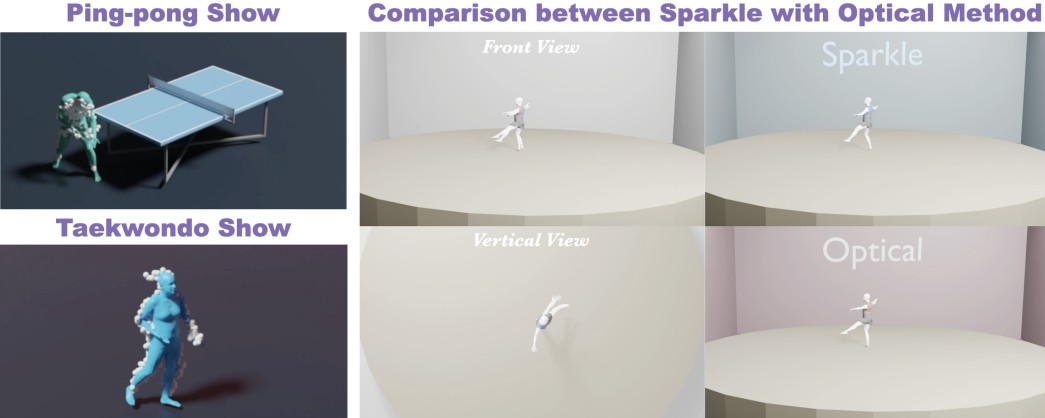

Figure 7: More sports demo shows and comparison with optical-based MoCap System.

## E.1 EVALUATION IN THE REAL WORLD

Our method, *SparkleMotion*, demonstrates exceptional robustness and generalizability across a wide spectrum of challenging real-world scenarios, directly validating the core contribution of the *Sparkle* representation—balancing expressiveness and robustness. As illustrated in Figure 1 and Figure 7, the framework seamlessly handles inputs from diverse sensor modalities, including sparse, long-range LiDAR point clouds and denser data from consumer-grade depth sensors like the iPhone. This sensor-agnostic performance stems from *Sparkle*'s factorized design, which allows it to maintain structural coherence even when the surface geometry from the point cloud is incomplete or noisy. This capability is critically evaluated in demanding situations such as basketball games with intense occlusions and large-scale soccer matches featuring over 20 players at distances beyond 80 meters. In these environments, where point clouds are highly fragmented, the explicit coupling of internal skeletal priors with external geometric evidence in *Sparkle* prevents error accumulation and ensures stable full-body motion estimation.

## E.2    DETAILS OF DEPLOYMENT

To validate the practical utility of *SparkleMotion*, we deployed it as a real-time system capable of operating in challenging, large-scale scenarios. Our complete network comprises **82.3 million parameters**. On a consumer-grade **NVIDIA GeForce RTX 4090** GPU, the system achieves a stable end-to-end frame rate of **60 FPS** in visualization by interpolation, the real frame rate is **10 FPS** limited in LiDAR throughput. This performance includes all stages of processing: point cloud preprocessing (e.g., Farthest Point Sampling), the core *Sparkle* estimation, and the final SMPL parameter regression. This efficiency demonstrates its suitability for real-world applications. Furthermore, the system exhibits strong scalability, qualitative results from large-scale scenes, such as soccer fields and basketball courts, are showcased in Figure 1.

## E.3    PERFORMANCE COMPARISON WITH OPTICAL SYSTEM

To quantitatively benchmark its accuracy against a gold standard, we compared *SparkleMotion* to an industry-grade OptiTrack optical motion capture system in a controlled laboratory setting. On a complex ballet sequence involving rapid spins and precise limb movements (Fig 7), our method achieved remarkable accuracy, with a mean Global MPJPE of 34.27 mm. This result confirms that the geometric initialization and refinement enabled by the *Sparkle* representation can yield precision comparable to marker-based systems for individual subjects. The significant advantage of our approach, however, lies in its scalability and accessibility. Unlike OptiTrack, which is confined to a calibrated volume, our system operates in large-scale, in-the-wild environments, requires no wearable markers or complex setup, and naturally supports an unrestricted number of subjects. These capabilities, showcased extensively in the supplementary video, underscore the potential of *SparkleMotion* as a versatile and practical solution for next-generation motion capture, bridging the gap between laboratory-grade accuracy and real-world applicability.

## E.4    MULTI-PERSON TRACKING AND SYSTEM SCALABILITY

While our core contribution is the *Sparkle* representation, we further demonstrate the practicality of the full system in large-scale, multi-person scenarios. Our end-to-end pipeline achieves a throughput of approximately 90 FPS on a single NVIDIA RTX 4090 GPU in football scenes with 25 persons, far exceeding the data capture rate of typical LiDAR sensors (e.g., 10 Hz). This confirms that the system introduces negligible latency.

A key design advantage is the decoupling of person segmentation and pose estimation. The pose estimation network processes each individual independently, ensuring that the **accuracy of the motion capture is invariant to the number of people** in the scene.

For robust person segmentation, we employ a clustering-based algorithm within a predefined activity zone, followed by a task-specific point cloud segmentation network. This offers superior generalization across diverse environments compared to object detectors trained for narrow domains. We benchmark our tracking pipeline against the strong baseline **ByteTrack** (Zhang et al., 2022) on the challenging **FIFA soccer dataset** (Jiang et al., 2024). As shown in Table E.4, our method achieves superior performance across standard multi-object tracking metrics, demonstrating more stable and accurate tracking.

To comprehensively evaluate tracking performance, we adopt several standard metrics:

- **MOTA** (Multiple Object Tracking Accuracy): An overall measure combining false positives, false negatives, and identity switches. Higher is better, with 1 being the ideal value.

- **IDF1** (ID F1-Score): Measures the correctness of identity preservation. Higher is better.

- **IDs** (Identity Switches): Counts how many times a tracked target loses its correct identity. Lower is better.

- **HOTA** (Higher Order Tracking Accuracy): A unified metric that balances detection and association accuracy. Higher is better.

| Method | MOTA ↑ | IDF1 ↑ | IDs ↓ | HOTA ↑ |
|--------|--------|--------|-------|--------|
| ByteTrack | 0.9485 | 0.7652 | 33 | 0.8451 |
| **Ours** | **0.9883** | **0.8172** | **18** | **0.9975** |

Table 12: Multi-person tracking performance comparison on the FIFA dataset.

## F    LIMITATIONS AND FUTURE WORK

While *SparkleMotion* demonstrates robust performance, it is important to discuss its limitations, which primarily stem from the final stage of the pipeline, the mapping of the learned *Sparkle* representation onto the parametric SMPL model. This dependency inherently restricts the framework to human subjects and bounds its expressivity within the topological and shape space of SMPL. Consequently, the model may struggle with highly unconventional poses, significant non-rigid deformations caused by loose clothing (e.g., long skirts or robes), or the detailed articulation of hands, which are beyond the representation capacity of the standard SMPL model. Furthermore, the front-end estimation of the *Sparkle* representation itself can be challenged under conditions of severe and persistent occlusion or extreme point cloud sparsity, where establishing reliable spatial correspondences becomes difficult.

These limitations, however, are not fundamental to the *Sparkle* representation itself but rather to its current instantiation within a human-specific, parametric modeling paradigm. This distinction opens several promising research directions. The most compelling future work lies in generalizing the core principle of *Sparkle*, the factorization of an object's state into internal structural keypoints and external surface anchors, beyond the human domain. We hypothesize that this representation serves as a powerful and universal inductive bias for learning from point clouds, applicable to a wide range of articulated and non-rigid objects, such as animals or robotic arms. Decoupling the *Sparkle* representation from SMPL could involve learning category-agnostic structural priors or employing an implicit surface decoder to reconstruct detailed geometry, thereby overcoming the current constraints on shape and deformation. Additional avenues include the development of more sophisticated temporal models to enhance long-term consistency and the exploration of self-supervised learning frameworks to reduce the reliance on extensive 3D annotations. In summary, while the current system is pragmatically limited by its parametric backbone, the underlying *Sparkle* representation offers a versatile and promising foundation for future research in general 3D dynamic understanding from sparse sensing.

## USAGE OF LARGE LANGUAGE MODELS

During the preparation of this work, the authors used LLM solely for the purpose of improving language fluency and checking grammatical errors in previously written drafts. After using this tool, the authors review and edit the content as needed and take full responsibility for the intellectual content of the publication.

