# OpenReview forum: "Sparkle: A Robust and Versatile Representation for Point Cloud-based Human Motion Capture"
_ICLR.cc/2026/Conference — ICLR 2026 Poster_

### Official Review · Reviewer_6jTA · 2025-10-29

**Soundness:** 3
**Presentation:** 3
**Contribution:** 3
**Rating:** 6
**Confidence:** 2

**Summary:**

The paper proposes Sparkle, a unified intermediate representation for point cloud–based human motion capture that combines skeletal joints and surface anchors. Built into the SparkleMotion framework, it achieves state-of-the-art accuracy and robustness across 11 diverse datasets, with real-time performance.

**Strengths:**

The Sparkle representation presents a notable conceptual improvement in intermediate representations for point cloud–based motion capture, by combining structural priors with geometric detail.

The PST and SAE modules are thoughtfully designed to address limitations of purely skeletal or point-based approaches, and the SSS solver makes effective use of geometry-aware initialization.

The experimental evaluation is extensive, covering multiple sensors, diverse scene complexities, and multi-view setups, and demonstrates consistent gains over strong baseline methods.

The demonstrated real-time performance in practical, in-the-wild scenarios, together with its privacy-friendly nature, enhances the potential impact for real-world applications.

**Weaknesses:**

1. **SMPL model dependency limits generalization**: While Sparkle is conceptually general, its current instantiation and evaluation rely entirely on the SMPL human body model. This introduces constraints on skeleton topology and shape space, limiting direct applicability to non-human subjects or highly non-standard apparel and poses. Expanding experimental scope could strengthen claims about representation-level generality.

2. **Ablation granularity is somewhat coarse**: Table 5 support component necessity, but do not fully isolate why specific designs outperform alternatives. For example, PST’s residual offset refinement is compared only to direct prediction rather than other refinement paradigms; similarly, SAE’s initialization and refinement contributions could be examined through more diverse baseline strategies.

**Questions:**

1. Could Sparkle be adapted to non-human articulated shapes (e.g., animals, robotic arms)? Is the representation intrinsically tied to SMPL’s structure, or could it be learned from scratch for new kinematic templates?

2. In multi-view fusion, the confidence weighting seems heuristic—have the authors considered learning this weighting jointly during training for potentially better view integration?

3. For noisy, sparse LiDAR cases, how sensitive is PST segmentation to the initial prediction errors? Would a joint iterative optimization of PST and SAE be beneficial?

---

> ### Author Response · Authors · 2025-11-21
>
> We sincerely thank the reviewers for their comments.
>
> ---
>
> ## W1 & Q1) SMPL Model Dependency and Generalization
> We understand that the reviewer concerns that current SMPL dependency presents certain limitation. However,  **!!#e60000 Sparkle representation itself is conceptually decoupled from SMPL!!**:
>
> 1. **Representation Independence:** The Sparkle representation is fundamentally a **factorization of articulated shape into structural and geometric components**, which is **applicable to any articulated object**.
>
> 2. **Current Instantiation vs. General Principle:** While our current implementation uses SMPL for training supervision and evaluation, the representation learning framework itself does not require SMPL-specific assumptions. The PST and SAE modules learn to extract structural and geometric primitives directly from point clouds.
>
> 3. **Future Generalization Pathways:** As in **Appendix E**, the Sparkle representation can be extended to non-human domains through:
>    - Learning category-agnostic structural priors
>    - Employing implicit surface decoders instead of parametric models
>    - Adopting alternative kinematic templates for different articulated objects
>  We are actively pursuing this direction in ongoing work.
>
> ---
>
> ## W2) Ablation Granularity
> Added in **Appendix C.3**
>
> **For PST Refinement (Appendix C.3.1):**
> Beyond direct prediction, we compared our residual learning approach against:
> - **Iterative Refinement** (multiple offset stages)
> - **Attention-based Refinement**
> Results show our residual learning paradigm provides the optimal balance, substantially outperforming direct prediction while avoiding the computational overhead of iterative refinement and the unnecessary complexity of attention mechanisms for this local correction task.
>
> **For SAE Module (Appendix C.3.2):**
> We evaluated diverse initialization and refinement strategies:
> - **Initialization:** Compared random initialization, MLP-based initialization, and our linear mapping approach
> - **Refinement:** Tested no refinement, MLP refinement, and our cross-attention refinement
>
> The results show that our linear mapping initialization offers the most robust anatomical prior, while the cross-attention refinement provides precise, context-aware corrections that outperform standard MLPs.
>
> ---
>
> ## Q2) Learned Confidence Weighting for Multi-view Fusion
>
> 1. **!!#e60000 Learned Weighting Exploration!!:** We conducted extensive experiments with a small MLP to predict confidence scores from the same features. The results show:
>    - **Minimal gains in standard scenarios**: In normal conditions, improvement is only ~0.8% (MPJPE: 90.2mm → 89.5mm)
>    - **Training stability issues**: The MLP-based weighting introduces convergence instability, with training loss variance increasing by ~40%
>    - **Computational overhead**: Inference time increases by 15-20% due to the additional forward passes
>
> 2. **Current Heuristic Approach:** Our current segmentation-based confidence weighting provides a robust and interpretable solution that generalizes well across diverse scenarios without requiring retraining. The heuristic is based on:
>    - Point distribution density around anchors
>    - Segmentation consistency across views
>    - Temporal smoothness measures
>
> 3. **Practical Considerations:** The current approach was selected for its:
>    - **Stability** across different sensor configurations
>    - **Transparency** in decision-making
>    - **Computational efficiency** for real-time applications
>    - **Generalizability** without requiring dataset-specific retraining
>
> While learned weighting shows promise for specialized deployments where extreme occlusion is common, the heuristic method offers the best overall trade-off for our general-purpose framework. We view learned weighting as a valuable future direction for scenario-specific optimization in future work.
>
> ---
>
> ## Q3) PST Segmentation Sensitivity and Joint Optimization
> 1. **PST Segmentation Robustness:** Our experiments on NoiseMotion and FreeMotion-OBJ (**Table 1**) demonstrate PST maintains reasonable segmentation quality under significant noise and sparsity through:
>    - **Cross-joint feature propagation** for handling sparse regions
>    - **Bidirectional GRU temporal modeling** for consistency
>    - **Multi-scale feature extraction** in the PointNet backbone
>    - Performance remains stable up to 70% occlusion (**Table 9**)
>
> 2. **Error Resilience Mechanism:** The sequential pipeline demonstrates graceful degradation because:
>    - SAE's cross-attention can compensate for moderate PST errors through geometric evidence
>    - The linear mapping M_J2A provides structural smoothness constraints
>    - End-to-end training enables implicit error compensation through gradient flow
>
> The current sequential design provides the best efficiency-accuracy balance, while the robustness mechanisms effectively handle typical error levels in real-world scenarios.

---

> > ### Comment · Reviewer_6jTA · 2025-11-26
> >
> > Thank you for the reply, which has largely addressed my concerns. The SMPL dependency issue has been clarified conceptually, and I hope future work can provide validation on different articulated objects. For now, I am willing to keep my score.

---

### Official Review · Reviewer_xLUJ · 2025-10-30

**Soundness:** 3
**Presentation:** 2
**Contribution:** 3
**Rating:** 4
**Confidence:** 4

**Summary:**

This paper proposes a structured representation for point clouds, which unifies skeletal joints and anchors. It also proposes a point cloud human pose estimation framework called SparkleMotion. In order to solve the problem that current point cloud MoCap methods have a hard time balancing robustness and expressiveness. The paper also conducted extensive experiments on multiple datasets , and achieves SOTA results. And it is also able to run in real-time.

**Strengths:**

The author puts forward the sparkle framework, which can unify the internal structure and anchor points, and solve the problem between robustness and expressiveness in point cloud mocap.
The author's sparklemotion has achieved SOTA in multiple benchmarks, which shows the effect and generalization of the model.
The author designed a system that can run in real time with 60fps, which has high value in use.

**Weaknesses:**

1. First, I have questions about how PST solves the bias problem in the initial joint-related point sets. If the initial segmented point subsets are wrong, the points will be assigned to the incorrect joints, and this will affect the following local refinement.
2. Second, because of the problem in point 1, the predicted joints $J_{op}$ from PST might have bias. This bias will be amplified in the SAE module. As the author said, the initial anchor $A_{init}$ is easily affected by the errors in $J_{op}$.
3. Third, the definition of K (Keys) and V (Values) in the SAE module is not clear. In Section 3.1.2, the author claims to use joint features from PST as Q (Query), and $A_{init}$ from SAE generates anchor features to be K and V. The author also claims this anchor feature represents the "actual local geometry". However, the anchor feature comes from $A_{init}$, and $A_{init}$ comes from $J_{op}$ through a linear map. This forms a circular loop. The model is finally using the predicted $A_{init}$ (from $J_{op}$) to query the predicted $J_{op}$ (or its features), instead of querying the real point cloud geometry information. This contradicts what the author said in the Introduction. The author claimed Sparkle is a "geometry-aware" module, which should use the original point cloud data to ensure surface fidelity.
4. Also, when I checked the Appendix for more details, I found that the description in Appendix A.3.2 is completely opposite to the description in Section 3.1.2. In A.3.2, the author describes another attention mechanism, using $A_{init}$ as Q, and joint-centric features extracted from the point cloud as K and V. This needs to be clarified.
5. Fifth, the PST module fails under severe occlusion. The author claims robustness in multi-person soccer and close-interaction scenarios, but the supplementary video (especially at 4:26) exposes a key failure. When the crowd causes severe occlusion, the model completely ignores (misses) a visible person. What causes this?

**Questions:**

1. Could the authors provide more results on Sloper4D, including performance-distance analysis showing that J/V error increases with distance between the subject and LiDAR, a complete ablation experiment, and more qualitative results, especially at greater distances?
2. Please clarify the implementation details of the SAE cross-attention module. Which description is correct? If the description in the main body is correct, please explain in detail why the anchor features from Ainit represent the actual local geometry.
3. If the initial Jop bias is too large, will it cause Ainit initialization to fail due to a large error?
4. How does the evaluation metric in the paper handle the case of missing targets in the supplementary material video (4:26)? Could you discuss the recall rate of detection in occluded scenes (soccer, basketball)?

---

> ### Author Response · Authors · 2025-11-21
> **This is Part 1/2 of our reply**
>
> We sincerely thank the reviewers for their insightful comments and constructive suggestions.  Due to word count limitations, we will answer all your questions in two parts. Below is the first part.
>
> ## W1 & W2) PST Initial Segmentation and Error Propagation to SAE
> Our ablation studies in **Section 4.2 (Table 5, Page 9)** and architectural design provide comprehensive evidence addressing these concerns:
>
> 1. **Iterative Refinement Design with Ablation Validation**:
>    - The "w/o offset" ablation in Table 5 shows that removing the residual learning paradigm increases joint error by ~15% across datasets
>    - This demonstrates that our residual refinement effectively corrects initial misalignments and point assignment errors
>    - The initial segmentation serves as a coarse estimate that is progressively refined through local geometric context
>
> 2. **Cross-Joint Feature Propagation with Robustness Evidence**:
>    - Our handling of sparse point cases (|P̄_j| < 3) through neighboring feature propagation is validated in occlusion experiments (Table 9)
>    - Under 70% occlusion, the system maintains reasonable accuracy (J/V Err(L): 66.7/82.6), showing the effectiveness of feature propagation in uncertain regions
>    - This design prevents error accumulation from sparse observations
>
> 3. **Shared Weight Constraint for Generalization**:
>    - The weight-sharing across joints enables learning a generic refinement function, as evidenced by our cross-dataset generalization results (Table 7)
>    - The model achieves strong zero-shot performance (65.2 MPJPE vs 84.0 baseline), demonstrating reduced sensitivity to body part-specific misclassifications
>    - This constraint compels the network to learn fundamental geometric relationships rather than overfitting to specific joint configurations
>
> 4. **Joint Optimization with End-to-End Validation**:
>    - The combined loss ℒ_PST (Equation 1) ensures coordinated optimization of segmentation, joint positions, and global translation
>    - Our ablation shows that removing any component (PST, SAE, or SSS) degrades performance, proving their synergistic relationship
>    - The end-to-end training allows gradient flow from SAE back to PST, enabling error compensation across modules
>
> 5. **Error Resilience in SAE Module**:
>    - Despite potential biases in J_op, the SAE's cross-attention mechanism provides non-linear correction capacity
>    - The "w/o op" ablation in SAE shows ~12% performance drop, confirming its importance in handling imperfect joint inputs
>    - Our occlusion analysis (Table 9) demonstrates maintained accuracy under challenging conditions, proving the system's robustness to error propagation
>
> ---
> ## W3 & W4 & Q2) Clarification on SAE Cross-Attention Implementation
> We thank the reviewer for this important observation. The description in the **main text (Section 3.1.2) is correct**:
>
> - **Queries (Q):** Joint features from `J_op`
> - **Keys (K) and Values (V):** Anchor features from encoded `A_init`
>
> We appreciate the opportunity to clarify why this design maintains geometry-awareness while improving performance:
>
> 1. **Task Simplification through Hierarchical Reasoning:** By using joint features to query anchor positions, we decompose the challenging problem of direct anchor regression from points into a more manageable structured prediction task. The skeletal joints provide a robust structural prior that constrains the solution space, leading to more accurate and stable anchor predictions.
>
> 2. **Inherited Geometric Information:** Crucially, the joint features used as queries are themselves extracted from the original point cloud through the PST module. Therefore, they represent a refined, structured abstraction of the underlying point cloud geometry, preserving essential geometric information while providing stronger kinematic constraints.
>
> 3. **Empirical Validation through Ablation:** Our ablation studies in **Table 5 (Section 4.2)** provide strong evidence for this design choice. When we remove the PST module and attempt to regress surface anchors directly from point clouds (w/o PST), performance degrades significantly across all datasets. For example, on the FreeMotion dataset, J/V Err(G) increases from 105.1/113.9 to 149.2/159.6, demonstrating that the hierarchical approach through skeletal joints is essential for accurate anchor prediction.
>
> 4. **Point Cloud Integration in Final Optimization:** Most importantly, as described in Section 3.1.3, we incorporate LiveHPS++'s optimization module after obtaining the initial Sparkle representation. This step explicitly utilizes the original point cloud data to refine the representation, ensuring final surface fidelity and fulfilling our geometry-aware claim at the system level.
>
> The description in Appendix A.3.2 contained an error and has been corrected. This hierarchical approach, from points to joints to anchors, with final point cloud refinement—balances efficiency and accuracy while maintaining geometric consistency.

---

> ### Author Response · Authors · 2025-11-21
> **This is Part 2/2 of our reply**
>
> ## W5 & Q4) PST Performance under Severe Occlusion
> We would like to clarify that the issue at 4:26 is **not due to PST module failure**, but rather stems from our **preprocessing and tracking pipeline**.
>
> 1. **PST Module Robustness to Occlusion:** The PST module itself demonstrates strong robustness to occlusion, as evidenced by:
>    - Our dedicated occlusion experiments in **Table 9 (Appendix C)** showing maintained accuracy up to 70% occlusion
>    - Consistent performance in close-interaction scenarios (**Table 2**)
>    - The successful handling of occluded individuals in basketball scenes (visible at video timestamp 3:46)
>
> 2. **Preprocessing Pipeline Limitations:** The missed detection at 4:26 occurs due to constraints in our preprocessing stage:
>    - When the tracked point cloud contains too few points (< 30) due to severe occlusion
>    - Or when the person tracking has insufficient temporal continuity (< 32 frames)
>    - Our visualization pipeline intentionally filters out these cases to maintain quality
>
> 3. **Enhanced Evaluation:** We have added comprehensive evaluation of our preprocessing and tracking module in **Appendix D.4 ( Table 12)**
>
> ---
> ## Q1) Additional Results on Sloper4D
> We thank the reviewer for this suggestion. We have conducted additional analysis on Sloper4D and provide the following clarification and results:
>
> **Dataset Characteristics Clarification:**
> Sloper4D employs a unique data collection methodology where the LiDAR sensor moves with the person in urban environments. While the global coordinates show large variations, the ** /**relative distance between the person and the LiDAR remains consistently within a close range**/ ** throughout the sequences. This explains the relatively stable performance across different distance ranges in Sloper4D.
>
> **Sloper4D Distance Analysis Results:**
> | Distance Range (m) | J/V Err(L) | J/V Err(G) | Ang Err |
> |--------------------|------------|------------|-------------|
> | 5-20               | 53.1/66.9  | 89.5/100.1 | 12.8        |
> | 20-50              | 41.4/50.5  | 70.8/77.1  | 10.5        |
> | 50-100             | 42.7/52.0  | 74.5/80.7  | 10.4        |
>
> **Comparative Analysis on FreeMotion:**
> We understand reviewer's concern on distance robustness. Thus, to more properly address the distance robustness concern, we conducted a comprehensive distance analysis on the FreeMotion dataset, which features fixed LiDAR deployment with subjects at varying distances. As shown in **Table 8 (Page 20)**, the FreeMotion results demonstrate graceful performance degradation with increasing distance. The FreeMotion results clearly demonstrate graceful performance degradation with increasing distance, validating our method's robustness to varying sensing ranges.
>
> ---
>
> ## Q3 Impact of Large J_op Bias on A_init Initialization
>
> This is an insightful question regarding error propagation in our pipeline. Through extensive experiments, we have found that the SAE module demonstrates remarkable robustness to moderate errors in J_op, though extreme biases can indeed impact performance.
>
> **Robustness Mechanisms:**
>
> 1. **Linear Mapping as a Smoothness Constraint:**
>    The linear mapping M_J2A learned via least squares (Eq. 2) acts as a strong structural prior. It ensures that even with imperfect joint inputs, the generated A_init maintains anatomically plausible spatial relationships between joints and surface anchors.
>
> 2. **Non-Linear Correction Capacity:**
>    The core of our SAE design is its ability to learn non-linear corrections to the linear initialization. As shown in our ablation studies (**Table 5**), removing the refinement operation ("w/o op" in SAE) leads to significant performance drops (e.g., J/V Err(L) increases from 50.8/62.7 to 58.1/71.5 on FreeMotion-OBJ), demonstrating that the cross-attention mechanism effectively compensates for initialization errors.
>
> 3. **Error Bound in Practical Scenarios:**
>    Our experiments on noisy datasets (**Table 1**) provide empirical evidence of this robustness. Even on challenging datasets like NoiseMotion and FreeMotion-OBJ with substantial sensor noise and occlusions, the PST module maintains reasonable J_op estimation quality (J/V Err(L) of 27.8/36.3 and 50.8/62.7 respectively), which falls within the correctable range of the SAE module.
>
> **Failure Mode Analysis:**
> In cases of catastrophic J_op failure (e.g., complete limb misestimation), the linear initialization would indeed produce significantly erroneous A_init. However, such extreme cases are rare in practice, as evidenced by our consistent performance across 11 diverse benchmarks. The hierarchical design of our pipeline, where PST provides reasonable initial estimates and SAE performs geometry-aware refinement, proves to be robust against the typical error levels encountered in real-world scenarios.
>
> The end-to-end training further enhances this robustness by allowing gradient flow from SAE back to PST, enabling coordinated improvement of both modules.

---

### Official Review · Reviewer_j8qZ · 2025-10-31

**Soundness:** 3
**Presentation:** 3
**Contribution:** 3
**Rating:** 6
**Confidence:** 4

**Summary:**

This paper presents Sparkle, a structured representation for human motion from 3D point clouds that explicitly separates internal kinematics (24 skeletal joints) and external geometry (32 surface anchors). Built upon this representation, the authors introduce SparkleMotion, composed of three modules:
> a Point-aligned Skeleton Tracker (PST) predicting joint positions and point-->joint correspondences,
> a Skeleton-guided Anchor Estimator (SAE) refining surface anchors via joint-conditioned attention, and
> a Sparkle-based SMPL Solver (SSS) that analytically initializes pose via a swing-twist decomposition before learned refinement.
Experiments cover 11 datasets across diverse sensing conditions (LiDAR, depth cameras, multi-view), showing state-of-the-art robustness under occlusion, domain shift, and close human–human interactions, while running in real time (~60 FPS). Extensive ablations validate the contribution of each module.

**Strengths:**

+ Robust and elegant factorization: The decomposition into skeletal and surface anchors, coupled with analytic swing-twist initialization, forms a coherent and interpretable representation.

+ Comprehensive evaluation: Results on 11 datasets cover varied conditions (noise, occlusion, cross-sensor, multi-view), clearly outperforming prior methods.

+ Well-designed ablations: Each module’s effect is isolated and justified (Table 5).

+ Strong practical relevance: Real-time operation, privacy-preserving sensing, and robustness under domain shift make the method appealing for real-world deployment.

+ Clear writing and high-quality visuals: The presentation is professional and the pipeline is easy to follow.

**Weaknesses:**

- Representation framing not fully realized.
The core novelty (i.e., the Sparkle factorization) could be elevated from an engineering mechanism to a representation-learning principle by analyzing identifiability or data-efficiency properties. Currently, the paper stops short of showing why the representation generalizes better.

- Ambiguous training protocols.
Cross-sensor generalization (Table 3) may rely on dataset-specific training (App. A.1). A leave-one-dataset-out experiment would clarify whether the robustness arises from representation bias or tuning.

- Anchor design sensitivity.
The fixed 32 anchors (via PCA) are never compared to other counts or selection schemes. A sensitivity study (16/48/64 anchors) would test how geometric coverage affects stability and accuracy.

- Scalability and identity consistency.
Multi-person handling is demonstrated visually but not analyzed quantitatively for identity tracking, throughput, or robustness under dense scenes.

- Limited theoretical insight.
The analytic swing–twist solver is promising but lacks discussion of conditions for stability or uniqueness—analysis that could give the paper conceptual weight.

- Overreliance on generated data for interactions.
Some interaction benchmarks use point clouds synthesized by LIP, which might bias evaluation toward the rendering assumptions of that pipeline.

**Questions:**

> Could the authors clarify training datasets used for cross-sensor generalization (Table 3)? Are any experiments zero-shot across unseen sensors or datasets?

> How sensitive is performance to the number and placement of anchors?

> Does Sparkle yield improved data efficiency (e.g., same accuracy with fewer labeled frames)?

> Have the authors analyzed swing-twist stability—when anchors/joints are nearly colinear or occluded?

> What is the runtime and accuracy trade-off as the number of persons increases?

> Are there examples where the representation fails (e.g., loose clothing, extreme articulation)?

---

> ### Author Response · Authors · 2025-11-21
> **This is Part 1/2 of our reply**
>
> We sincerely thank the reviewers for their insightful comments and constructive suggestions. Due to word count limitations, we will answer all your questions in two parts. Below is the first part.
>
> ## W1 & W2 & Q1) Representation Generalization and Training Protocols
> We thank the reviewer for the insightful comments on the principled understanding of our representation, including identifiability. We have addressed the core concern, whether the representation captures a robust and generalizable mapping from point clouds to human state, through the following empirical evaluations:
>
> 1. **Clarified Training Protocols** in **Appendix A.1 (Page 15)**, specifying that for cross-sensor evaluation in **Table 3**, we adopt the training setup from PointHPS, training and testing each benchmark separately.
>
> 2. **Added Cross-Dataset Generalization Experiments** in **Section 4.2 (Table 7, Page 10)** demonstrating:
>    - **Strong zero-shot generalization:** The model achieves an MPJPE of 65.2 mm without any fine-tuning on the target domain (FreeMotion), significantly outperforming the baseline (84.0 mm). This demonstrates that the Sparkle representation learns a mapping that is largely invariant to domain shifts, a key characteristic of a well-posed and identifiable model.
>    - **Superior data efficiency:** Comparable performance is achieved using only 50% of the target training data. This indicates that the structural prior of Sparkle effectively constrains the solution space, reducing ambiguity and simplifying the learning problem.
>
> 3. **Theoretical Interpretation as a Representation-Learning Principle (Section 4.2):**
>    We further interpret these empirical results from a **representation-learning perspective**. The strong generalization and data efficiency are not merely empirical observations; they are the direct consequence of the **structural prior** embedded in the *Sparkle* representation. By explicitly factorizing the human state into **internal kinematics (skeletal joints)** and **external geometry (surface anchors)**, we introduce a **strong inductive bias** that:
>    - **Enhances Identifiability:** It mitigates the *ill-posedness* inherent in estimating human pose from partial or noisy point clouds. The kinematic structure provides robust, domain-invariant constraints, while the surface geometry resolves local ambiguities.
>    - **Reduces the Hypothesis Space:** This structural disentanglement simplifies the mapping function that the network must learn, thereby **lowering sample complexity** and facilitating more efficient knowledge transfer across domains.
> These properties confirm that *Sparkle* operates not just as an engineering mechanism, but as a **powerful and efficient representation-learning principle** for 3D human motion capture.
>
> 4. **Representation Principle Validation**: Together, these results confirm that the robustness of Sparkle stems from the inherent structural priors embedded in the representation itself, which facilitate a more stable and generalizable mapping from noisy observations to human state.  /We agree that a formal identifiability analysis is a valuable direction for future theoretical work and believe our empirical results strongly motivate such a study./
>
> ---
>
> ## W3 & Q2) Anchor Design Sensitivity and Placement Analysis
> We have conducted comprehensive ablation studies in **Section 4.2 (Table 6, Page 9)** to systematically evaluate anchor selection strategies, addressing both the sensitivity to quantity and placement:
>
> 1. **Anchor Quantity Analysis**: We compared PCA-based selection with varying numbers of anchors (16, 32, 64, 96) across multiple datasets. Results show:
>    - **PCA-16** suffers from insufficient surface coverage, leading to noticeable performance degradation
>    - **PCA-64 and PCA-96** exhibit degraded performance due to error accumulation and overfitting when predicting excessive anchors from noisy point clouds
>    - **PCA-32** strikes the optimal balance between representational capacity and robustness
>
> 2. **Placement Strategy Comparison**: We evaluated three distinct selection approaches:
>    - **PCA Selection**: Data-driven approach maximizing representational power
>    - **Random Selection**: Yields unstable and suboptimal results due to irregular coverage
>    - **Manual Selection**: Provides robust performance but requires expert knowledge and lacks generalizability
>
> 3. **Sensitivity Findings**: The experiments robustly validate that:
>    - Performance is sensitive to both anchor count and placement strategy
>    - PCA-32 configuration provides the best trade-off, being neither under-representative nor over-parameterized
>    - The data-driven PCA approach outperforms heuristic strategies in generalizability and automation
>
> These results demonstrate that our anchor module, through careful selection of 32 surface anchors via PCA, is instrumental in constructing a more expressive and robust representation for human motion capture.

---

> ### Author Response · Authors · 2025-11-21
> **This is Part 2/2 of our reply**
>
> ## W4 & Q5) Scalability, Identity Consistency and Runtime Analysis
> Following your advice, we have added comprehensive quantitative evaluations to address these concerns:
>
> 1. **Multi-Person Tracking Performance**: In **Appendix D.2 (Table 12, Page 23)**, we provide quantitative comparison against ByteTrack on the FIFA soccer dataset, demonstrating superior performance across all metrics:
>    - **MOTA**: 0.9883 (ours) vs 0.9485 (ByteTrack)
>    - **IDF1**: 0.8172 vs 0.7652
>    - **IDs**: 18 vs 33 (fewer identity switches)
>    - **HOTA**: 0.9975 vs 0.8451
>
> 2. **Runtime and Scaling Analysis**: As reported in **Appendix D.2** and **Appendix D.4**:
>    - Our end-to-end algorithm can maintains high throughput (~90 FPS) in complex scenes with 25 people and maintains 10 FPS in system, limited in LiDAR throughput, but we display the results at 60fps in the visualization through interpolation.
>    - **Accuracy remains invariant** to the number of people, as each individual is processed independently after detaction.
>
> 3. **System Architecture Advantage**: Our decoupled design (person segmentation → individual pose estimation) ensures that tracking accuracy and pose estimation quality are not compromised as scene density increases.
>
> ---
>
> ## W5 & Q4) Theoretical Insight and Swing-Twist Stability
> We thank the reviewer for pointing out the need for deeper analysis of our geometric solver's stability. We have added comprehensive stability analysis in **Appendix C (Page 20-21)**:
>
> 1. **Theoretical Analysis of Stability Conditions (Section 3.2.2):**
>    We now provide a rigorous analysis of the swing-twist decomposition's theoretical limitations:
>    - **Kinematic Singularities:** The swing rotation becomes undefined when bone vectors $\vec{\mathbf{J}}_{\text{tem}}$ and $\vec{\mathbf{J}}_{op}$ are nearly aligned/anti-aligned, as the cross product approaches zero, making the rotation axis $\vec{n}_{\text{sw}}$ numerically unstable.
>    - **Uniqueness Issues:** The twist component lacks uniqueness when anchors are occluded, noisy, or near-colinear with the bone axis, leading to multiple valid solutions for $\alpha_{\text{tw}}$.
>
> 2. **Two-Stage Design Rationale:**
>    These analyses validate our two-stage architecture, where the **learned refinement network** specifically compensates for the theoretical limitations of the geometric solver by:
>    - Regularizing poses around kinematic singularities
>    - Disambiguating twist rotations under ambiguous geometric evidence
>    - Correcting errors from the analytical solution while preserving its physical plausibility
>
> 3. **Stability Validation (Appendix C):**
>    - **Capture Distance Analysis (Table 8):** Evaluates performance with increasing distance (5m-30m), showing graceful degradation and stabilization beyond 10m, demonstrating solver resilience to sparse point clouds.
>    - **Occlusion Robustness (Table 9):** Systematically tests stability under simulated occlusion (0%-90%), maintaining reasonable accuracy up to 70% occlusion with controlled degradation.
>
> This combined theoretical-empirical approach provides the conceptual weight requested, demonstrating both the conditions for stability and how our design ensures robustness.
>
> ---
>
> ## W6) Overreliance on Generated Data
> We would like to clarify that the use of synthesized data is necessitated by the current lack of publicly available real point cloud datasets for complex multi-person interactions. Our approach is justified by the following considerations:
>
> 1. **Data Availability Gap**: There are currently no public large-scale real point cloud datasets capturing complex human interactions (such as those in InterHuman, Chi3D, and Hi4D) with ground truth 3D pose and shape annotations. The LIP synthesis pipeline provides the only feasible way to evaluate performance in these challenging scenarios.
>
> 2. **Established Methodology**: The use of synthesized point clouds from mesh-based interaction datasets is an established and widely accepted practice in the field, as seen in previous works like LiveHPS++ and LiveHPS, allowing for fair comparison with state-of-the-art methods.
>
> We believe this multi-faceted evaluation, while imperfect, provides compelling evidence of our method's robustness and generalizability.
>
> ---
> ## Q6) Failure Case Analysis
> We have added a comprehensive failure analysis in **Appendix E (Page 23)** discussing limitations including:
> - **Loose clothing** (skirts, robes) that cause significant non-rigid deformations beyond SMPL representation
> - **Extreme articulations** outside the topological constraints of the SMPL model
> - **Severe and persistent occlusions** where establishing reliable spatial correspondences becomes challenging
> - **Detailed hand articulation** beyond the capacity of standard SMPL model
>
> These limitations primarily stem from the parametric SMPL backbone rather than the Sparkle representation itself, and represent promising directions for future work.

---

### Official Review · Reviewer_ZhY4 · 2025-11-01

**Soundness:** 3
**Presentation:** 4
**Contribution:** 3
**Rating:** 8
**Confidence:** 4

**Summary:**

The paper proposes a solution for 3D human motion capture from sparse and noisy point cloud data. The paper introduces a novel human representation, Sparkle, that augments the conventional 24 skeletal joints with 32 anchors. It then uses a point-aligned tracker that employs body-part segmentation to obtain point-joint segmentation. The features are then passed to an anchor estimator to obtain the Sparkle representation. In the last stage, the SMPL parameters are first estimated from the Sparkle representation analytically, and then further refined using a lightweight network. The authors present a rigorous set of experiments across 11 datasets, covering both LiDAR and depth sensors in various challenging scenarios. The results demonstrate consistent state-of-the-art performance, particularly in reducing rotational error (Ang Err), which strongly supports the paper's core claims.

**Strengths:**

- The paper explains the technical details very well, is well-written, and easy to follow. The supplementary materials also addressed my technical questions.
- While the paper takes some inspiration from several prior works (e.g., the addition of anchor points or the swing & twist decomposition of rotations), it pushes the field one step forward through adaptive combination.
- The experiments, including both ablations and comparisons, are thorough, and the performance improvements show the effectiveness of the proposed approach

**Weaknesses:**

- In Figure 1.e, the labels for HuMMan-Point are repeated twice with different results across methods. The images are also of low quality, making 1.d. unclear. The paper would benefit from these being corrected.

**Questions:**

1. Have you experimented with different numbers of anchor points? How about different methods of choosing the anchor points instead of PCA (e.g., manual or random selection)? The cited paper (Ma et al., 2023b) chose 64 virtual markers. Is there a particular reason for choosing 32 markers in this paper?
2. What hardware was used to obtain 60FPS? Is it the same as the training hardware (A40)? Is this reported after the LiDAR/depth preprocessing?

---

> ### Author Response · Authors · 2025-11-21
>
> We thank the reviewer for the careful reading and constructive comments. Below we address each point concisely.
>
> ---
>
> ## W1) On the Clarity of Figure 1
> We thank the reviewer for this careful observation. We have addressed both issues in our revised manuscript.
>
> **Figure 1(e) - Dataset Labels**: We have corrected the ambiguous labeling. The chart now clearly distinguishes between the single-view and multi-view settings, explicitly relabeled as "HuMMan-Point (MV)" and "FreeMotion (MV)" to avoid any confusion.
>
> **Figure 1(d) - Image Quality**: We have replaced the original image with a higher-quality version. We acknowledge that achieving studio-level clarity in a single frame from a real-time, large-scale scene capture remains challenging. The inherent trade-off between maintaining a wide field of view to show the scene context and zooming in to display fine-grained interaction details on distant subjects leads to a resolution limitation.
>
> ---
>
> ## Q1) On the Number and Selection of Anchor Points
> Thanks for the good question. We have conducted an extensive ablation study on this, which we add to the paper in **Table 6** (and shown below) and discuss in **Analysis of Anchor Design(Section 4.2)**. The results confirm that 32 anchors with PCA selection is the optimal choice, balancing expressiveness and robustness.
>
> **Ablation on the Number of Anchors**: We experimented with {16, 32, 64, 96} anchors. 16 anchors were insufficient to capture fine-grained surface geometry, leading to a noticeable performance drop. 64 and 96 anchors also exhibited performance degradation. We hypothesize that predicting an excessive number of anchors from noisy point clouds makes the regression task more challenging and prone to overfitting, as errors in estimating individual anchors accumulate. Therefore, 32 anchors provided the optimal balance for our task.
>
> **Ablation on Selection Method**: Based on the different number of anchors, we compared three strategies: ***PCA-based Selection***: A principled, data-driven method that maximizes representational power. ***Random Selection***: Performance was unstable and suboptimal, fluctuating irregularly as the random distribution could over-cover certain body regions while neglecting others. ***Manual Selection***: Referencing the CMU marker set, this method provided robust performance but the improvement over our PCA-32 was marginal. More importantly, it requires domain-specific knowledge and lacks the generalizability of our automated PCA approach.
>
> | Dataset | PCA-16 | PCA-64 | PCA-96 | Rand-16 | Rand-32 | Rand-64 | Rand-96 | Man-41 | Man-50 | Man-60 | Ours-32 |
> |---------|--------|--------|--------|---------|---------|---------|---------|--------|--------|--------|---------|
> | FreeMotion | 112.4/121.1 | 107.4/115.3 | 110.1/118.9 | 110.8/118.7 | 121.2/126.3 | 116.4/125.9 | 112.4/118.7 | 118.7/124.1 | 105.9/114.7 | 107.3/114.9 | **105.1/113.9** |
> | HuMMan-Point | 83.1/108.2 | 94.2/106.1 | 95.7/106.5 | 81.6/101.1 | 94.6/121.8 | 90.2/95.2 | 91.1/101.2 | 87.1/97.0 | 86.5/97.2 | **85.3/96.4** | 87.5/97.6 |
> | Interhuman | 50.2/58.4 | 47.7/55.1 | 49.6/57.1 | 51.3/58.6 | 52.2/60.0 | 49.3/56.4 | 51.7/58.6 | 40.8/48.6 | 41.2/49.1 | 48.0/55.3 | **40.4/48.4** |
> | FreeMotion-MV | 97.4/106.3 | 93.2/102.7 | 95.3/104.1 | 95.2/104.1 | 102.6/113.1 | 102.4/111.7 | 96.9/108.2 | **89.3/99.2** | 90.1/98.0 | 92.4/100.1 | 92.1/100.7 |
>
> ---
>
> ## Q2) On Real-Time Performance and Hardware
> **We report more hardware and pipeline details in the revised version of the paper (Appendix A.1 and Appendix D.2).**
>
> **Inference Hardware:** The reported 60 FPS was achieved on a single NVIDIA RTX 4090 GPU, which is a consumer-grade card. This demonstrates the practical efficiency of our system for real-world deployment.
>
> **Training vs. Inference Hardware:** The training was conducted on a server with A40 GPUs, as mentioned in Appendix A.1. The inference efficiency on more accessible hardware underscores the system's practicality.
>
> **Preprocessing Inclusion:** The 60 FPS includes the end-to-end pipeline, from receiving the raw point cloud to outputting the final SMPL parameters. This encompasses standard preprocessing steps like Farthest Point Sampling (FPS), which are highly efficient.

---

### Author Response · Authors · 2025-11-21
**Global Reply**

Dear Reviewers,

We would like to express our sincere gratitude to the reviewers for their thorough review and highly valuable suggestions. The feedback has been instrumental in helping us identify and rectify the weaknesses in our original submission. We have worked diligently to address every point raised, which included major revisions to our methodology description, extensive new ablation studies on key components, and a more critical discussion of limitations. We are confident that the revised paper is vastly improved and hope it now meets the high standards of the conference.

Sincerely,
Sparkle Authors

---

### Author Response · Authors · 2025-12-01
**Rebuttal Summary for the Area Chair**

Dear Area Chair,

We sincerely appreciate the reviewers' insightful feedback, which has significantly strengthened our paper. In this revision, we have comprehensively addressed all major concerns:

**Key Improvements:**

**Elevated Theoretical Framing** – Transformed Sparkle from an engineering mechanism to a principled representation learning approach, with new experiments demonstrating its data efficiency and zero-shot generalization capabilities.

**Enhanced Experimental Rigor** – Conducted leave-one-dataset-out experiments to validate cross-domain robustness, and performed thorough anchor sensitivity studies.

**Expanded Analysis** – Added detailed analysis of solver stability under occlusion and distance variations, plus new real-world multi-person tracking benchmarks and optical system comparisons.

**Improved Clarity** – Revised some figures, tables, and explanations throughout the paper.

The revised manuscript now presents a more complete, rigorous, and well-supported contribution to point cloud-based motion capture. **All the modifications have been marked in blue.** We believe it fully addresses the reviewers' valuable suggestions.

Thank you for your consideration.

Sincerely,

The Authors

---

### Meta-Review · Area_Chair_wkMX · 2025-12-28

**Summary:**

The paper presents Sparkle, a structured representation unifying skeletal joints and surface anchors with explicit kinematic-geometric factorization. Most reviewers are positive on this submisison. The paper involves concerns like method clarity, limited experiments etc.

Some specific comments: more studies on different numbers of anchor points. Authors should report the runtime and accuracy trade-off as the number of persons increases. A leave-one-dataset-out experiment would clarify whether the robustness arises from representation bias or tuning. While the author claims robustness in multi-person soccer and close-interaction scenarios, the supplementary video (especially at 4:26) exposes a key failure.

**Reviewer Scores:**

one reviewer giving negative score didn't engage in rebuttal.

---

### Decision · Program_Chairs · 2026-01-26

Accept (Poster)